# Rapid DNA replication origin licensing protects stem cell pluripotency

Jacob Peter Matson[1], Raluca Dumitru[2], Philip Coryell[3], Ryan M Baxley[4], Weili Chen[5], Kirk Twaroski[5], Beau R Webber[5], Jakub Tolar[5], Anja-Katrin Bielinsky[4], Jeremy E Purvis[3,6], Jeanette Gowen Cook[1,6]*

[1]Department of Biochemistry and Biophysics, The University of North Carolina, Chapel Hill, United States; [2]Human Pluripotent Stem Cell Core Facility, The University of North Carolina, Chapel Hill, United States; [3]Department of Genetics, The University of North Carolina, Chapel Hill, United States; [4]Department of Biochemistry, Molecular Biology, and Biophysics, The University of Minnesota, Minneapolis, United States; [5]Stem Cell Institute, University of Minnesota, Minnesota, United States; [6]Lineberger Comprehensive Cancer Center, University of North Carolina, Chapel Hill, United States

**Abstract** Complete and robust human genome duplication requires loading minichromosome maintenance (MCM) helicase complexes at many DNA replication origins, an essential process termed origin licensing. Licensing is restricted to G1 phase of the cell cycle, but G1 length varies widely among cell types. Using quantitative single-cell analyses, we found that pluripotent stem cells with naturally short G1 phases load MCM much faster than their isogenic differentiated counterparts with long G1 phases. During the earliest stages of differentiation toward all lineages, MCM loading slows concurrently with G1 lengthening, revealing developmental control of MCM loading. In contrast, ectopic Cyclin E overproduction uncouples short G1 from fast MCM loading. Rapid licensing in stem cells is caused by accumulation of the MCM loading protein, Cdt1. Prematurely slowing MCM loading in pluripotent cells not only lengthens G1 but also accelerates differentiation. Thus, rapid origin licensing is an intrinsic characteristic of stem cells that contributes to pluripotency maintenance.

DOI: https://doi.org/10.7554/eLife.30473.001

*For correspondence:
jean_cook@med.unc.edu

Competing interests: The authors declare that no competing interests exist.

## Introduction

Metazoan DNA replication requires initiation at thousands of DNA replication origins during S phase of every cell cycle. Origins are genomic loci where DNA helicases first unwind DNA and DNA synthesis begins. Origins are made competent for replication during G1 phase of each cell cycle by the loading of minichromosome maintenance (MCM) complexes onto DNA. MCM is the core component of the replicative helicase, and the process of MCM loading is termed origin licensing. Total MCM levels remain constant throughout the cell cycle, but the levels of DNA-loaded MCM change as cells progress through the cell cycle. Cells can begin MCM loading as early as telophase and loading continues throughout G1 until the G1/S transition, the point of maximum DNA-loaded MCM (*Kimura et al., 1994*; *Todorov et al., 1995*). Throughout S phase, individual MCM complexes are activated for DNA unwinding as origins 'fire'. MCM complexes travel with replication forks and are progressively unloaded as replication forks terminate (*Figure 1a*) (*Deegan and Diffley, 2016*; *Remus and Diffley, 2009*; *Siddiqui et al., 2013*).

The control of origin licensing is critical for genome stability. Origins must not be re-licensed after S phase begins because such re-licensing can cause a genotoxic phenomenon known as re-replication which may result in double strand breaks, gene amplification, aneuploidy, and general genome

**eLife digest** From red blood cells to nerve cells, animals' bodies contain many different types of specialized cells. These all begin as stem cells, which have the potential to divide and make more stem cells or to specialize.

All dividing cells must first unwind their DNA so that it can be copied. To achieve this, cells load DNA-unwinding enzymes called helicases onto their DNA during the part of the cell cycle known as G1 phase. Cells must load enough helicase enzymes to ensure that their DNA is copied completely and in time. Stem cells divide faster than their specialized descendants, and have a much shorter G1 phase too. Yet these cells still manage to load enough helicases to copy their DNA. Little is known about how the amount, rate and timing of helicase loading varies between cells that divide at different speeds.

Now Matson et al. have measured how quickly helicase enzymes are loaded onto DNA in individual human cells, including stem cells and specialized or "differentiated" cells. Stem cells loaded helicases rapidly to make up for the short time they spent in G1 phase, while differentiated cells loaded the enzymes more slowly. Measuring how the loading rate changed when stem cells were triggered to specialize showed that helicase loading slowed as the G1 phase got longer. Matson et al. found that the levels of key proteins required for helicase loading correlated with the rates of loading. Altering the levels of the proteins changed how quickly the enzymes were loaded and how the cells behaved – for example, slowing down the loading of helicases made the stem cells specialize quicker.

These findings show that the processes of cell differentiation and DNA replication are closely linked. This study and future ones will help scientists understand what is happening during early animal development, when specialization first takes place, as well as what has gone wrong in cancer cells, which also divide quickly. A better understanding of this process also helps in regenerative medicine – where one of the challenges is to make enough specialized cells to transplant into a patient with tissue damage without those cells becoming cancerous.

DOI: https://doi.org/10.7554/eLife.30473.002

instability (*Arias and Walter, 2007*; *Truong and Wu, 2011*). To avoid re-replication, MCM loading is tightly restricted to G1 phase by multiple overlapping mechanisms that destroy or inactivate MCM loading proteins to prevent any new origin licensing after S phase begins (*Remus and Diffley, 2009*; *Arias and Walter, 2007*; *Truong and Wu, 2011*). On the other hand, cells typically load 5- to 10-fold more MCM complexes in G1 than they strictly require under ideal circumstances, and the additional MCM loading ensures timely and complete genome duplication even if replication hurdles are encountered in S phase (*Ibarra et al., 2008*; *Woodward et al., 2006*; *Ge et al., 2007*). It is possible for cells to proliferate with less than optimal MCM loading, but such cells are hypersensitive to DNA damage and replication stress (*Blow et al., 2011*; *McIntosh and Blow, 2012*).

MCM loading to license origins is restricted to G1, but G1 length varies widely among different cell types. For example, specialized developmental and immune cell cycles have minimal G1 lengths of mere minutes (*O'Farrell et al., 2004*; *Kinjyo et al., 2015*; *Kermi et al., 2017*). In cultured human embryonic stem cells, G1 is only 2–3 hr, and this short G1 is both a hallmark of and has been implicated in maintaining pluripotency (*Soufi and Dalton, 2016*; *Kareta et al., 2015a*). G1 lengthens early in differentiation, and in cultured somatic cells is often greater than 12 hr (*Calder et al., 2013*). Thus, different proliferating cells have drastically different amounts of time available to complete MCM loading before making the G1-to-S phase transition. In addition, pluripotent stem cells respond to differentiation stimuli specifically in G1 phase, suggesting that the balance among cell cycle phases influences differentiation potential (*Gonzales et al., 2015*; *Pauklin and Vallier, 2013*).

Given that MCM loading is restricted to G1 and the wide variation of G1 lengths, we postulated that the absolute amount of loaded MCM in S phase is a product of both the time spent in G1 and the rate of MCM loading. The combination of these two parameters has implications for genome stability because loading more or less MCM in G1 influences S phase length and how effectively S phase cells can accommodate both endogenous and exogenous sources of replication stress (*Shima et al., 2007*; *Pruitt et al., 2007*). These implications are relevant both when cell cycle

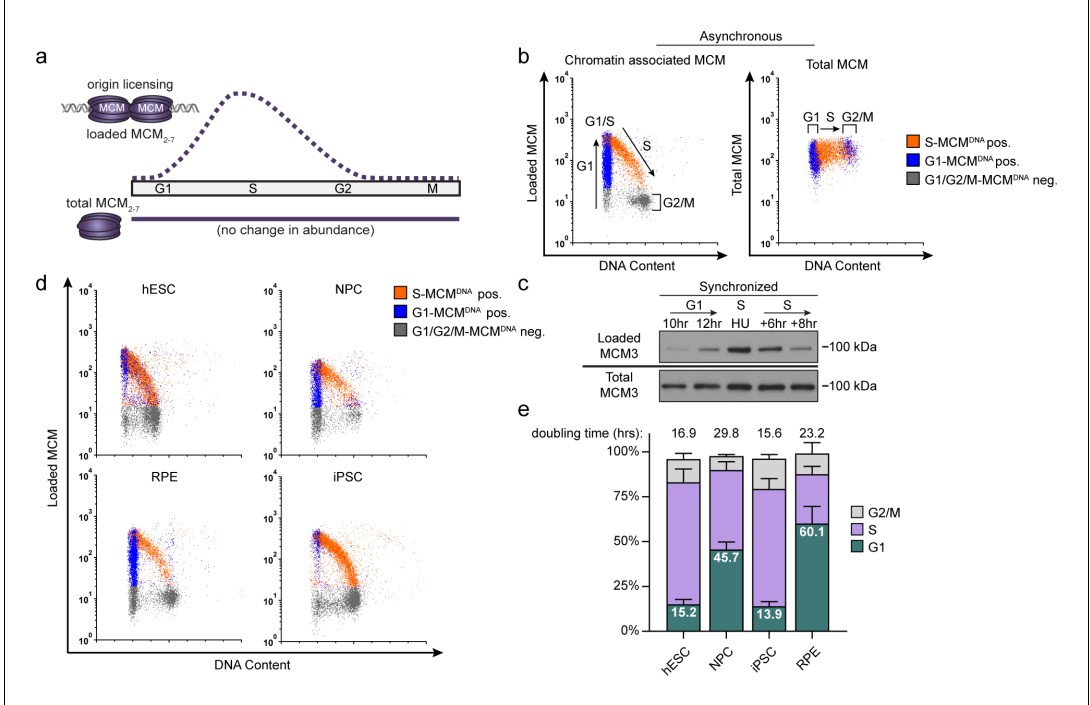

**Figure 1.** Pluripotent stem cells load MCMs faster than differentiated cells. (**a**) DNA-loaded MCM levels increase in G1 and decrease in S phase, whereas total MCM protein levels are constant throughout the cell cycle. (**b**) Flow cytometric analysis of DNA-loaded and total MCM in asynchronously proliferating RPE1-hTERT cells. Cell cycle phases are defined by DNA content and DNA synthesis. Left: Cells were labeled with EdU, extracted with nonionic detergent to remove unbound MCM, fixed, and stained with anti-MCM2 (a marker for the MCM$_{2-7}$ complex), DAPI (total DNA), and for EdU incorporation (active DNA synthesis). Orange cells are S-phase with DNA-loaded MCM, blue cells are G1-phase with DNA-loaded MCM, and grey cells are G1/G2/M phase cells without DNA-loaded MCM. Right: Cells were treated as on the left except that they were fixed prior to extraction to detect total MCM2. (**c**) T98G cells were synchronized in G0 by contact inhibition and serum deprivation, then released into G1 for 10 or 12 hr, or re-synchronized in early S with hydroxyurea (HU), and released into S for 6 or 8 hr. MCM3 in chromatin-enriched fractions (Loaded) or whole cell lysates (Total) was detected by immunoblotting. (**d**) Chromatin flow cytometry of the indicated asynchronous cell lines measuring DNA content (DAPI), DNA synthesis (EdU incorporation), and loaded MCM (anti-MCM2). Blue Cells are G1-MCM[DNA]-positive and EdU-negative, orange are S phase-MCM[DNA]-positive; grey are G1/G2/M-MCM[DNA]-negative. (**e**) Stacked bar graph of cell cycle phase distribution from cells in (**d**); mean with error bars ± SD (n = 3 biological replicates). The percentage of G1 cells in each population is reported in the green sectors. The doubling times were calculated experimentally using regression analysis in GraphPad Prism.

DOI: https://doi.org/10.7554/eLife.30473.003

The following figure supplements are available for figure 1:

**Figure supplement 1.** Flow cytometry gating.

DOI: https://doi.org/10.7554/eLife.30473.004

**Figure supplement 2.** Validation of chromatin flow cytometry.

DOI: https://doi.org/10.7554/eLife.30473.005

**Figure supplement 3.** Characterization of pluripotent and differentiated cells.

DOI: https://doi.org/10.7554/eLife.30473.006

distributions change during development and during oncogenesis since many cancer cell lines also have short G1 phases. The actual rate of MCM loading in human cells has not yet been quantified, however, and little is known about how the amount, rate or timing of MCM loading varies between cells with different G1 lengths. Here, we utilized single-cell flow cytometry to measure MCM loading rates in asynchronous populations of pluripotent and differentiated cells. We discovered that rapid MCM loading is intrinsic to pluripotency, slows universally during differentiation, and rapid replication licensing suppresses differentiation. These findings demonstrate that the rate of MCM loading is subject to developmental regulation, and we suggest that rapid origin licensing is a new hallmark of pluripotency.

## Results

### Pluripotent cells load MCM significantly faster than differentiated cells

We considered two possibilities for how cells with varying G1 lengths load MCM onto DNA. One possibility is that cells with short G1 phases load MCM at the same rate as cells with long G1 phases resulting in less total loaded MCM. Alternatively, cells with short G1 phases could load MCM faster than cells with long G1 phases and reach similar levels of loaded MCM. To distinguish between these scenarios, we developed an assay to measure DNA-loaded MCM in individual cells of asynchronously proliferating populations by adapting a previously reported flow cytometry method (*Håland et al., 2015*; *Moreno et al., 2016*). We extracted immortalized epithelial cells (RPE1-hTERT) with nonionic detergent to remove soluble MCM. We then fixed the remaining chromatin-bound proteins for immunofluorescence with anti-MCM2 antibody as a marker of the $MCM_{2-7}$ complex, and for DNA content (DAPI) and DNA synthesis (EdU) to measure cell cycle phases (*Figure 1b*, left). We used a sample without primary antibody as a control (relevant flow cytometry gating schemes are shown in *Figure 1—figure supplement 1*). For chromatin flow cytometry, MCM signal below the antibody threshold is colored grey ('G1/G2/M-MCM$^{DNA}$ neg'), whereas detectable MCM signal is colored either blue in G1 cells ('G1-MCM$^{DNA}$ pos') or orange in S phase cells ('S-MCM$^{DNA}$ pos'). As expected, total MCM protein levels do not substantially change during the cell cycle (*Figure 1b*, right) (*Todorov et al., 1995*). For comparison to commonly used cell fractionation methods to assess MCM dynamics, we also probed immunoblots of chromatin-enriched fractions, and noted a similar MCM expression, G1 loading, and S phase unloading pattern (*Figure 1c*) (*Cook et al., 2002*; *Méndez and Stillman, 2000*). Interestingly, individual G1 cells (blue stripe) have a very broad range of DNA-loaded MCM levels with a more than 100-fold difference between minimum and maximum (*Figure 1b*, left). MCMs are unloaded during S phase, ending in G2/M with undetectable MCM on DNA (*Figure 1b*, left). Moreover, loaded MCM is resistant to extraction in high-salt buffer which removes peripherally bound chromatin proteins (*Figure 1—figure supplement 2f–h*), similar to yeast MCM complexes loaded *in vitro* (*Bowers et al., 2004*; *Randell et al., 2006*). We validated MCM2 antibody specificity using quiescent G0 synchronized cells (MCM unloaded), and we also observed the same broad G1 signal distribution using MCM3 antibody (*Figure 1—figure supplement 2a–d*).

Loaded MCM complexes are extremely stable on DNA, both in vivo and in vitro (*Cocker et al., 1996*; *Evrin et al., 2009*; *Remus et al., 2009*; *Bowers et al., 2004*). In human cells, MCMs can persist on DNA for more than 24 hr during a cell cycle arrest and are typically only unloaded during S phase (*Kuipers et al., 2011*). These properties result in MCM loading that occurs unidirectionally throughout G1 phase (*Symeonidou et al., 2013*). The unidirectional nature of MCM loading means that G1 cells with low MCM levels are in early G1, and G1 cells with high MCM levels are in late G1. Since we observed a broad distribution of MCM loading throughout G1 including many cells with low levels of loaded MCM, we conclude that RPE1-hTERT cells load MCM relatively slowly during their ∼ 9 hr G1.

We then used this method to analyze MCM loading in asynchronous cells with different G1 lengths. H9 human embryonic stem cells (hESCs) have a short G1 phase and spend most of the cell cycle in S phase. In contrast to the differentiated epithelial cells, the majority (∼80%) of G1 hESCs had high levels of loaded MCM; very few G1 cells had low levels of loaded MCM (blue cells, *Figure 1d*). This difference suggests that hESCs load MCM rapidly to achieve abundant DNA-loaded MCM in a short time. To test if MCM loading varies in differentiated cells, we differentiated hESCs into neural progenitor cells (NPCs) to generate an isogenic pair of pluripotent and differentiated cells. In contrast to hESCs, differentiated NPCs had a longer doubling time and a wide distribution of DNA-loaded MCM in G1 (blue cells, *Figure 1d*); they also spend approximately five time longer in G1 (*Figure 1e*; e.g. 15% of a 17 hr hESC cell cycle is 2.5 hr in G1 vs 45% of a 30 hr NPC cell cycle is 13.6 hr in G1). Since the NPCs had many cells with low levels of DNA-loaded MCM, we conclude that these differentiated cells load MCM more slowly than hESCs.

To generate another isogenic pair of pluripotent and differentiated cells, we reprogramed ARPE-19 primary retinal pigmented epithelial cells (RPE) into induced pluripotent stem cells (iPSCs). The iPSCs had hallmark features of pluripotency as measured by microscopy, bisulfite sequencing, gene expression, and teratoma formation (*Figure 1—figure supplement 3*), and their G1 phases were typically seven times shorter than their differentiated parents (*Figure 1e*). Like hESCs, the

pluripotent iPSCs had predominantly high levels of DNA-loaded MCM in G1 (*Figure 1d*). Importantly, both of the pluripotent cell lines reached approximately equal levels of DNA-loaded MCM at the start of S phase as their differentiated counterparts did, but in in less time (the absolute MCM loading intensities are comparable when samples are processed and analyzed with identical instrument settings). Taken together, these data demonstrate that pluripotent cells load MCMs rapidly in G1, but differentiated cells load MCMs slowly.

We then quantified the relative MCM loading rates in pluripotent and differentiated cells using ergodic rate analysis, a mathematical method that can derive rates from fixed, steady state populations (*Kafri et al., 2013*). Ergodic analysis can measure any unidirectional rate parameter from a steady state distribution and is not limited to the cell cycle (e.g. car traffic jams) (*Gray and Griffeath, 2001*). The ergodic analysis as applied to the cell cycle means that within a steady state population with a constant doubling time and cell cycle distribution, the number of cells at any point in the cell cycle is inversely related to the rate they move through that point. For any measured parameter, the density of cells indicates rate: low cell density on a flow cytometry plot indicates a fast rate passing through that cell cycle state, whereas high cell density indicates a slow rate. This phenomenon is analogous to a high density of slow-moving cars observed at a given point on a road in a traffic jam compared to a low density of fast-moving cars on an open highway. We visualized MCM loading as histograms of the MCM$^{DNA}$ intensities in only the G1 cells for ergodic rate analysis (G1-MCM$^{DNA}$, *Figure 2a,b* and *Figure 2—figure supplement 1*).

To compute MCM loading rate per hour, we experimentally determined the cell cycle distributions and doubling times of each cell population (*Figure 2—figure supplement 1*). Pluripotent cells reached near equal levels of loaded MCM at the G1/S transition in less time than differentiated cells. To quantify the actual MCM loading rate difference, we subdivided the G1-MCM$^{DNA}$ population into 10 equally-sized bins, calculated the MCM loading rate for each bin, then the overall average MCM loading rate for each G1 population. These calculations revealed that pluripotent hESCs loaded MCM 6.5 times faster per hour than differentiated NPCs and pluripotent iPSCs loaded MCM 3.9 times faster per hour than differentiated RPEs (*Figure 2c*). Thus, pluripotent cells with short G1 phases load MCMs significantly faster than differentiated cells with long G1 phases.

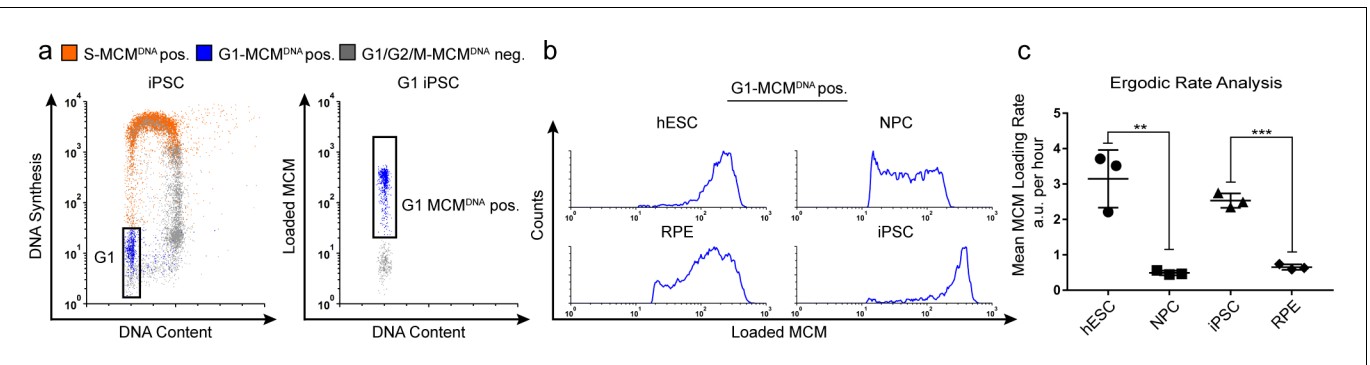

**Figure 2.** Quantification of MCM loading rate by ergodic rate analysis. (a) Gating scheme for chromatin flow cytometry of iPSCs measuring DNA content (DAPI), DNA synthesis (EdU incorporation), and loaded MCM (anti-MCM2); this sample is from *Figure 1d*. (b) Histograms of only the G1-MCM$^{DNA}$-positive cells from the four chromatin flow cytometry samples in *Figure 1d*. (c) Calculated mean MCM loading rate per hour by ergodic rate analysis; mean with error bars ± SD. (n = 3 biological replicates), unpaired two tailed t-test. **p=0.0049. ***p=0.001. See Materials and methods for details. See also *Figure 2—source data 1*.

DOI: https://doi.org/10.7554/eLife.30473.007

The following source data and figure supplement are available for figure 2:

**Source data 1.** Raw ERA values.
DOI: https://doi.org/10.7554/eLife.30473.009
**Figure supplement 1.** Ergodic rate analysis binning.
DOI: https://doi.org/10.7554/eLife.30473.008

## Differentiation, G1 length, and MCM loading rate are coupled

We hypothesized that MCM loading is fundamentally linked to pluripotency because MCM loading rate decreased during differentiation and increased during reprogramming. This idea predicts that slowed MCM loading is a phenomenon common to differentiation towards all germ layers. To test that hypothesis, we initiated differentiation in hESCs toward the three embryonic germ layers (neuro-ectoderm, mesoderm and endoderm), collecting cells at 24 hr and 48 hr after inducing differentiation (*Figure 3—figure supplement 1*). We confirmed progress toward each lineage by the expected gene expression changes, particularly induction of lineage-specific markers and modest reduction of pluripotency markers – even at these very early time points (*Figure 3c*). We assessed MCM loading rates during differentiation by flow cytometry as before. The MCM loading rate clearly decreased for all germ layers rapidly within the first 48 hr of initiating differentiation (*Figure 3a*, compare the grey histograms for undifferentiated G1 cells to the green and blue histograms). The decrease in MCM loading rate also coincided with the increase in the proportion of G1 cells for each lineage. For example, within 24 hr of neuroectoderm differentiation, G1 had already lengthened and MCM loading had slowed, but during mesoderm (BMP4) differentiation both G1 lengthening and slowed MCM loading took 48 hr (*Figure 3a and b*). The closely coordinated changes that we universally observed during differentiation suggest that MCM loading rate is coupled to G1 length. Importantly, these results demonstrate that the rate of origin licensing by MCM loading is developmentally regulated.

We next asked if G1 length and MCM loading rate are *obligatorily* coupled, or if the link can be short-circuited by artificially advancing the G1/S transition. To distinguish between these possibilities, we constructed an RPE1-hTERT derivative with a *CYCLIN E1* cDNA under doxycycline-inducible control. Cyclin E1 overproduction reproducibly shortened G1 length, consistent with previous studies (*Figure 4a,b*) (*Resnitzky et al., 1994*; *Ekholm-Reed et al., 2004*). Strikingly, cells overproducing Cyclin E1 (designated as '↑Cyclin E1') not only spent less time in G1 but also began S phase with much lower amounts of loaded MCM compared to the control; this new subpopulation appeared in the central triangular region of the plots that is typically clear of S phase cells (*Figure 4c,d* orange S-MCM$^{DNA}$pos). Cyclin E1 overproduction dramatically increased the proportion of these MCM-low early S phase cells by sixfold from 9.9% of control S phase cells to 63.6% of ↑Cyclin E1, S phase cells (*Figure 4c-e*). We also conclude that the MCM loading rate did not increase to accommodate the shorter G1 because the MCM loading pattern in G1 remained constant and the ↑Cyclin E1 cells had on average at least two-fold less DNA-loaded MCM in early S phase than control cells (*Figure 4f–h*). Furthermore, the ↑Cyclin E, MCM-low cells incorporated significantly less EdU per unit time than MCM-high cells did (1.6 fold lower mean, 1.8 fold lower median), indicating that low levels of loaded MCM are insufficient for normal S phase progression (*Figure 4i*). The early S phase cells with the least MCM loaded also had the least DNA synthesis by EdU intensity (data not shown). Thus, shortening G1 length without increasing MCM loading rate causes G1 cells to enter S phase prematurely without the full complement of DNA-loaded MCM.

Previous studies have shown that CDKs can inhibit MCM loading by directly inhibiting MCM loading factors, such as by stimulating Cdt1 degradation (Cdc10-dependent transcript 1), a protein essential for MCM loading (*Ekholm-Reed et al., 2004*; *Sugimoto et al., 2004*; *Tanaka and Diffley, 2002*). Cdt1 levels in lysates of asynchronous cells indeed decreased upon Cyclin E1 overproduction (*Figure 4—figure supplement 1a*). On the other hand, since Cdt1 is stable in G1 phase and degraded in S phase, the lower Cdt1 signal could have reflected less Cdt1 in the lysate due to the higher proportion of S phase cells; this indirect effect could apply to any cell cycle-regulated protein in cell populations with different cell cycle distributions. To test that idea, we measured total Cdt1 protein levels specifically in G1 by flow cytometry (*Figure 4—figure supplement 1b*). Cyclin E overproduction did not significantly reduce Cdt1 G1 levels relative to control (1.1-fold higher mean, 1.2 higher median, *Figure 4—figure supplement 1c*). We validated the Cdt1 antibody for immunofluorescence flow cytometry (*Figure 4—figure supplement 1d–f*). We conclude that Cyclin E/Cdk2 inhibits MCM loading indirectly, at least in part, by shortening G1 and decreasing the time available for MCM loading.

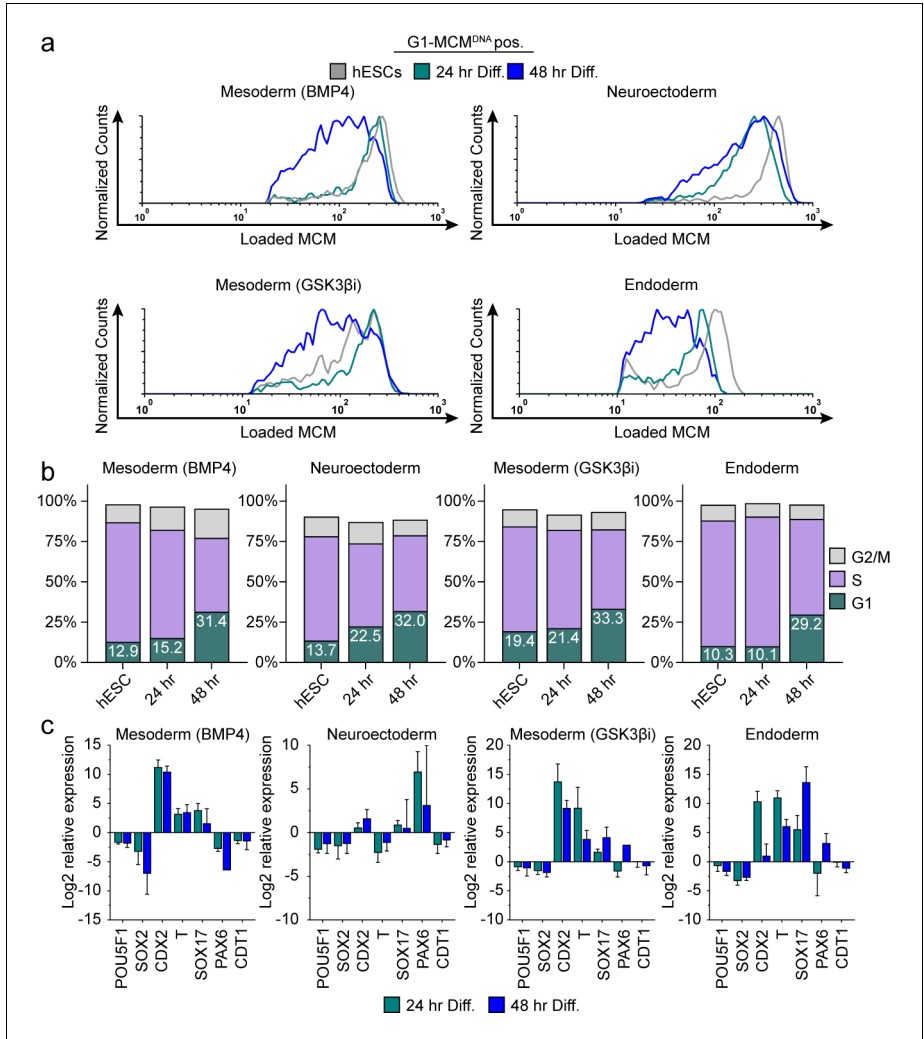

**Figure 3.** Differentiation universally decreases MCM loading rate. (**a**) Chromatin flow cytometry of hESCs induced to differentiate toward mesoderm (BMP4), neuroectoderm, mesoderm (GSK3βi), or endoderm for 24 or 48 hr. Histograms show only G1-MCM$^{DNA}$ cells positive as in *Figure 2b*. See methods for differentiation protocols. Cell counts for 24 hr and 48 hr were normalized relative to corresponding hESC samples. (**b**) Stacked bar graphs of cell cycle distribution for cells in (**a**). (**c**) Gene expression analysis of differentiation markers by quantitative PCR of the samples in (**a**); log$_2$ expression is relative to the undifferentiated cells. Data are mean ±SD of two biological replicates.

DOI: https://doi.org/10.7554/eLife.30473.010

The following figure supplement is available for figure 3:

**Figure supplement 1.** Stem cell differentiation.
DOI: https://doi.org/10.7554/eLife.30473.011

## Fast loading hESCs have more Cdt1 in G1

Loading MCM complexes onto DNA requires the six subunit Origin Recognition Complex (ORC), Cdc6, and Cdt1. We hypothesized that fast MCM loading in pluripotent stem cells is achieved by increased levels of the loading proteins. To test this idea, we probed protein lysates of asynchronous cells to compare the amount of MCM loading proteins between isogenic cell lines. Total Mcm2 and ORC protein levels remained constant (*Figure 5a*). The other MCM loading factors normally change in their abundance during the cell cycle due to regulated proteolysis. Cdc6 protein levels are low in G1 and high in S phase (*Figure 5b*). Conversely, Cdt1 protein levels are high in G1 and low in S phase (*Figure 5b*) (*Mailand and Diffley, 2005*; *Pozo and Cook, 2016*). Since an asynchronous

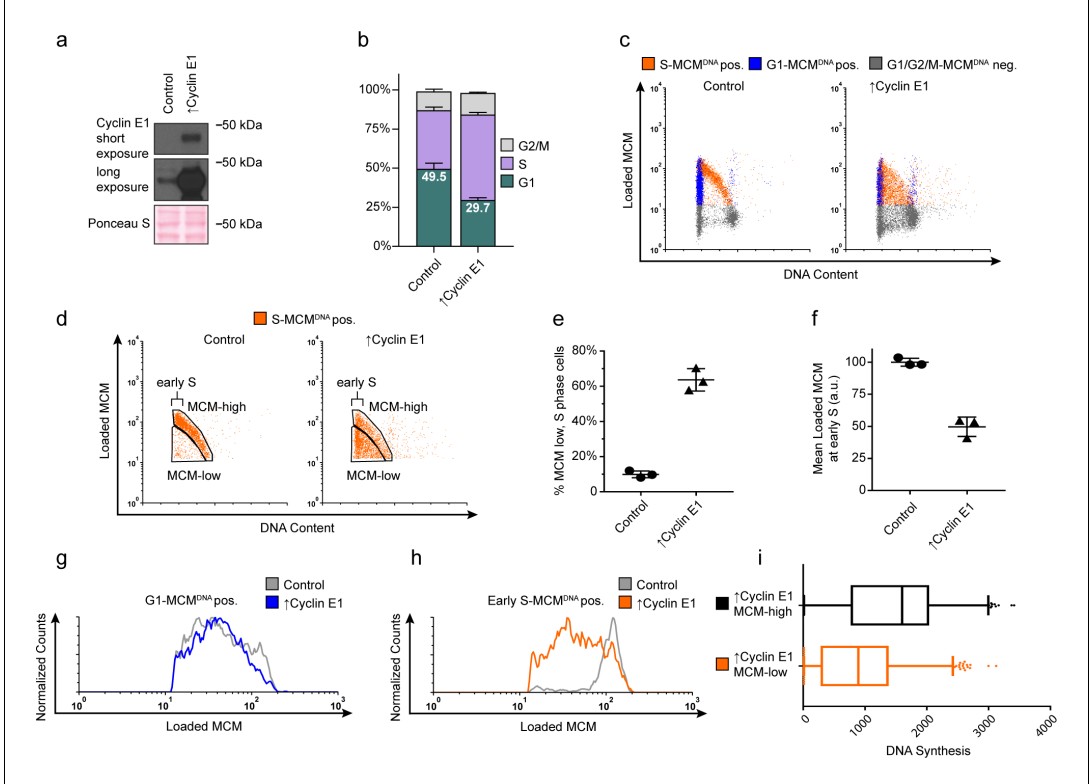

**Figure 4.** Cyclin E overproduction uncouples MCM loading and G1 length. (**a**) Immunoblots of a stable derivative of RPE1-hTert cells bearing an integrated doxycycline-inducible *cyclin E* construct treated with 100 ng/mL doxycycline for 72 hr to overproduce Cyclin E1 (↑Cyclin E1) or with vehicle control. (**b**) Stacked bar graphs of cell cycle distribution measured by flow cytometry for cells shown in (**a**); mean with error bars ± SD (n = 3 biological replicates). (**c**) Chromatin flow cytometry of control or cyclin E-overproducing cells measuring DNA content (DAPI), DNA synthesis (EdU incorporation), and loaded MCM (anti-MCM2). (**d**) S phase-MCM^DNA-positive cells from samples in (**c**) divided into populations that began S phase with high or low MCM^DNA. Early S cells are S phase cells with G1 DNA content. (**e**) The percentage of MCM^DNA positive, but-low MCM signal intensity S phase cells out of all S-MCM^DNA-positive cells from three biological replicates; mean with error bars ± SD, unpaired two tailed t-test. ***p=0.002. (**f**) Mean loaded MCM in early S phase, (S-MCM^DNA positive, G1 DNA content) from three biological replicates; mean with error bars ± SD, unpaired two tailed t-test. ***p=0.0004. (**g**) Histogram of G1-MCM^DNA-positive cells from samples shown in (**c**). Counts for ↑Cyclin E1 are normalized to the control. (**h**) Histogram of early S cells from samples shown in (**d**). Counts for ↑Cyclin E1 are normalized to the control. (These data are one of the replicates quantified in (**f**).). (**i**) EdU intensity from ↑Cyclin E1, MCM-high or MCM-low cells from (**d**) as box-and-whiskers plots. Center line is median, outer box edges are 25th and 75th percentile, whiskers edges are 1st and 99th percentile, individual data points are lowest and highest 1%, respectively. Median EdU incorporation of MCM-high ↑Cyclin E1 cells is 1.8 fold greater than MCM-low, mean EdU incorporation is 1.6-fold greater in MCM-high than MCM-low, average of three biological replicates. Samples compared by unpaired, two tailed t-test, **p=0.0027, **p=0.0033, respectively.
DOI: https://doi.org/10.7554/eLife.30473.012

The following figure supplement is available for figure 4:

**Figure supplement 1.** G1 Cdt1 levels are unaffected by Cyclin E overproduction.
DOI: https://doi.org/10.7554/eLife.30473.013

population of pluripotent cells spends significantly more time in S phase than differentiated cells do, we expected Cdc6 levels to be higher in asynchronous pluripotent cells compared to their isogenic counterparts. Cdc6 was indeed higher in pluripotent cells, as was Geminin, a protein regulated in a similar manner as Cdc6 (*Figure 5a*) (*McGarry and Kirschner, 1998*). To our surprise, even though the majority of asynchronous pluripotent cells were in S phase, a time when Cdt1 is degraded, Cdt1 levels were higher in asynchronous pluripotent cells than in isogenic differentiated cells (*Figure 5a*). A similar observation was reported for mouse embryonic stem cells (*Ballabeni et al., 2011*). These data imply that Cdt1 levels are higher in G1 phase of pluripotent cells than G1 of differentiated cells, providing a potential explanation for fast MCM loading in pluripotent cells.

To directly measure Cdt1 levels specifically in G1, we collected asynchronous hESCs and NPCs then fixed and stained them for Cdt1, EdU and DAPI. S phase Cdt1 degradation in hESCs is similar

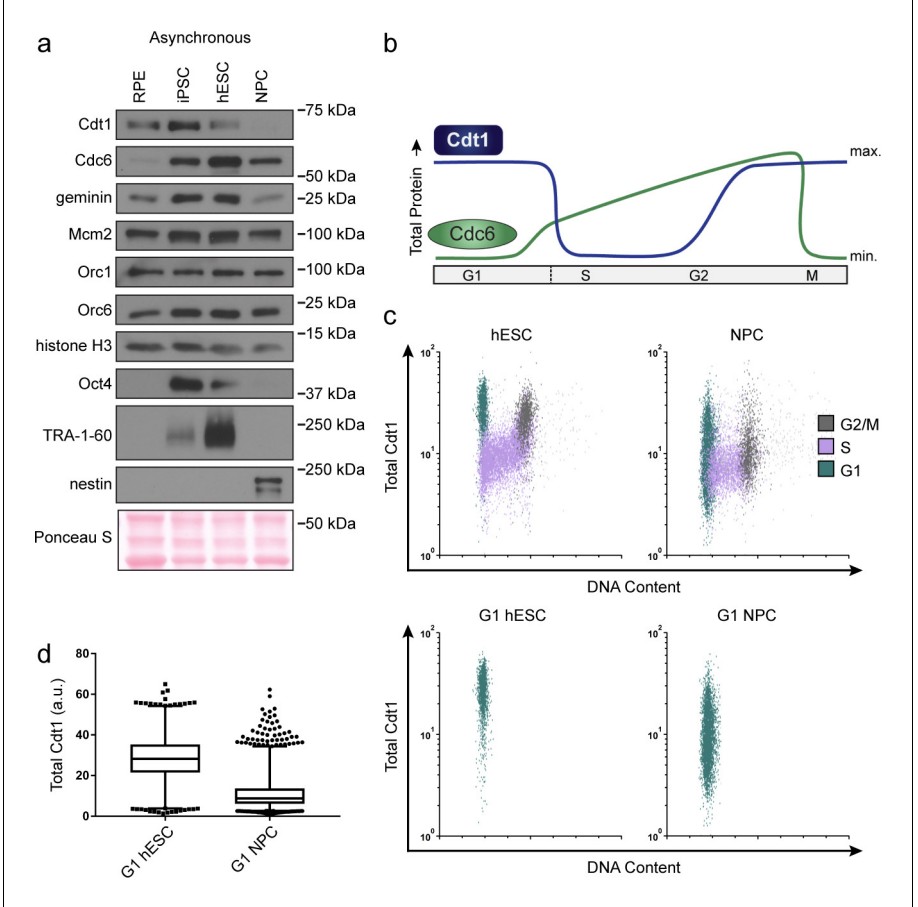

**Figure 5.** hESCs have high levels of Cdt1 in G1. (a) Immunoblots of whole cell lysates from the indicated asynchronous cell lines. (b) Expected changes in total protein levels of Cdt1 and Cdc6 during the human cell cycle. (c) Total Cdt1 detected in asynchronous cells by flow cytometry measuring DNA content (DAPI), DNA synthesis (EdU incorporation), and Cdt1 (anti-Cdt1). Green cells are G1, purple cells are S phase (EdU positive), grey cells are G2/M. (d) Box-and-whiskers plots of G1 Cdt1 concentration per cell from (C). Center line is median, outer box edges are 25th and 75th percentile, whiskers edges are 1st and 99th percentile, individual data points are lowest and highest 1%, respectively. Median G1 Cdt1 in hESCs is 2.9-fold greater, mean is 2.2-fold greater than G1 Cdt1 in NPCs, mean p=0.0504 median p=0.0243, average of three biological replicates. Flow plots are G1 cells only (green) from (c).

DOI: https://doi.org/10.7554/eLife.30473.014

to differentiated cells with very low levels in S phase (purple, *Figure 5c*). In contrast, the hESCs had a large population of cells with high Cdt1 levels in G1 (green cells) and significant amounts of Cdt1 in G2/M phase (grey cells), whereas the NPCs had a broad and overall lower Cdt1 distribution in G1 and very little Cdt1 in G2/M (*Figure 5d*). The G1 hESCs consistently harbored significantly more Cdt1 than G1 NPCs did (2.9-fold higher median, 2.2-fold higher mean, three replicates *[Figure 5d]*). We note that *CDT1* mRNA is modestly but consistently higher in asynchronous hESCs compared to differentiated derivatives, and that Cdt1 protein levels decrease during early differentiation coincident with the slowdown in licensing rate, but before loss of Oct4 (*Figure 3c* and *Figure 3—figure supplement 1*). We postulate that the higher amount of this essential MCM loading protein specifically in G1 contributes to the fast MCM loading rate in hESCs.

## Manipulating MCM loading factors alters MCM loading rates

Cdt1 is essential for MCM loading (*Pozo and Cook, 2016*); therefore, reducing Cdt1 levels should slow MCM loading. If MCM loading rate is linked to G1 length, then slowing MCM loading by reducing Cdt1 levels could also lengthen G1. To test this prediction, we used siRNA to reduce Cdt1 in

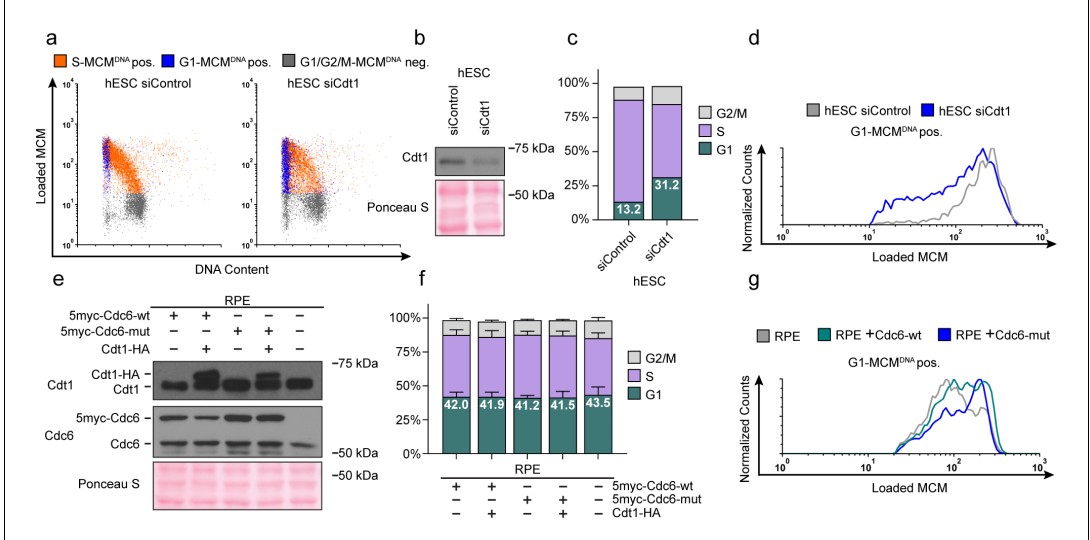

**Figure 6.** Manipulating MCM loading factors alters MCM loading rates. (**a**) Chromatin flow cytometry for hESCs treated with 25 nM siCdt1 or 100 nM siControl for 24 hr and labeled with EdU for 30 min prior to harvest. (**b**) Immunoblot of total protein from cells in (**a**). (**c**) Stacked bar graph of cell cycle distributions for samples in (**a**); representative of two biological replicates. The percentage of G1 cells in each population is reported in the green sectors. (**d**) Histograms of loaded MCM in G1-MCM^DNA cells. Counts for siCdt1 are normalized to the corresponding siControl sample. (**e**) Immunoblots of Cdt1 and Cdc6 in RPE cells with combinations of the following: constitutive production of 5Myc-Cdc6-wt or 5myc-Cdc6-mut (not targeted for degradation by APC^CDH1: R56A, L59A, K81A, E82A, N83A) and an integrated doxycycline-inducible Cdt1-HA construct treated with 100 ng/mL doxycycline for 48 hr to overproduce Cdt1-HA. (**f**) Stacked bar graphs of cell cycle distribution measured by flow cytometry for cells shown in (**e**); mean with error bars ± SD (n = 3 biological replicates). The percentage of G1 cells in each population is reported in the green sectors. (**g**) Histogram of loaded MCM in G1-MCM^DNA-positive cells from (**e**). Counts of Cdc6-wt and Cdc6-mut are normalized to parent RPE controls.

DOI: https://doi.org/10.7554/eLife.30473.015

The following figure supplement is available for figure 6:

**Figure supplement 1.** Manipulating MCM loading factors alters MCM loading rates.

DOI: https://doi.org/10.7554/eLife.30473.016

hESCs and measured changes in both MCM loading rate and G1 length (*Figure 6a,b*). As expected, Cdt1 depletion reduced MCM loading rate in hESCs (*Figure 6c*). Strikingly, G1 length increased coincidentally with the decrease in MCM loading rate (*Figure 6d*). These data corroborate the close link between MCM loading rate and G1 length in hESCs.

As a complement to slowing MCM loading in hESCs with a short G1, we also attempted to accelerate origin licensing in cells with a long G1 by overproducing essential licensing proteins. We first constructed an RPE_hTERT derivative with ~two-fold inducible Cdt1 overproduction, but this manipulation was insufficient to accelerate MCM loading (*Figure 6—figure supplement 1b*). We also tested ectopic myc-tagged Cdc6 expressed constitutively in RPE cells (*Figure 6e*); this addition also had only minimal effects on MCM loading rate (*Figure 6g*, compare the grey and green histograms). We considered, however, that human Cdc6 is unstable throughout much of G1 phase because it is targeted for degradation by APC^Cdh1 *Mailand and Diffley, 2005*; *Petersen et al., 2000*. We therefore expressed a previously-described Cdc6 mutant that is resistant to APC^CDH1-mediated destruction both alone and combination with inducible Cdt1 (*Figure 6e* and *Figure 6—figure supplement 1a*) (*Mailand and Diffley, 2005*; *Petersen et al., 2000*). We have previously demonstrated that tagged Cdc6 and Cdt1 are functional (*Cook et al., 2002*; *Coleman et al., 2015*; *Chandrasekaran et al., 2011*). Expression of the stable Cdc6-mut was sufficient to increase MCM loading rates (*Figure 6g*, compare the blue histogram to the grey and green histograms); Cdt1 overproduction had little additive effect on MCM loading rate in RPE cells (*Figure 6—figure supplement 1b*). Interestingly accelerating MCM loading by this method did not shorten G1 in RPEs (*Figure 6f*), further demonstrating that the length of G1 phase and the rate of MCM loading to license origins can be uncoupled. Slow MCM loading may delay S phase entry through the licensing checkpoint

(*Teer et al., 2006*; *Shreeram et al., 2002*; *Nevis et al., 2009*; *Ge and Blow, 2009*), but rapid MCM loading itself is not sufficient to trigger S phase entry.

## Rapid MCM loading protects hESC pluripotency

Our demonstration that slower MCM loading occurs universally during early differentiation suggested a functional link between the rate of MCM loading and pluripotency maintenance. We considered that slowing MCM loading might promote differentiation. To explore this idea, we prematurely slowed MCM loading in hESCs by Cdt1 depletion prior to inducing their differentiation (*Figure 7e*). After Cdt1 depletion, we stimulated differentiation toward mesoderm with BMP4 (*Bernardo et al., 2011*). After 48 hr, we quantified Oct4 and Cdx2 by immunostaining (*Figure 7a*, *Figure 7—figure supplement 1*). The pluripotency transcription factor Oct4 and the homeobox transcription factor Cdx2 reciprocally repress one another's expression, creating a clear distinction between Oct4-positive Cdx2-negative pluripotent cells and Oct4-negative Cdx2-positive differentiating cells (*Niwa et al., 2005*). We quantified the mean fluorescence intensity of both Oct4 and Cdx2 in >18,000 cells per condition with a customized, automated CellProfiler pipeline, plotting the signal intensities for each cell in a density scatter plot (*Figure 7b,c*). Stimulating control hESCs with 10 ng/mL of BMP4 slightly shifted the population toward differentiation, but most cells remained pluripotent with high Oct4 levels at this time point. Strikingly, hESCs pretreated with Cdt1 siRNA to prematurely slow MCM loading gained a substantial population of Oct4 negative-Cdx2 positive differentiating cells relative to controls that were treated similarly. To quantify the extent of differentiation, we divided the Cdx2 intensity of each cell by its Oct4 intensity, creating a single differentiation score (*Figure 7d*). After 10 ng/ml BMP4 treatment, Cdt1-depleted hESCs had significantly higher scores, indicating that prematurely slowing MCM loading promoted differentiation (p<0.0001, two-tailed Mann-Whitney test). Both control cells and Cdt1-depleted cells differentiated more fully at a higher concentration of 50 ng/mL BMP4, but the Cdt1-depleted cells still differentiated further than the controls (p<0.0001, two-tailed Mann-Whitney test, *Figure 7b* and *Figure 7—figure supplement 2a*). Other combinations of BMP4 concentrations or treatment times also resulted in a consistent, significant increase in differentiation in cells pretreated to slow MCM loading (p<0.0001, two-tailed Mann-Whitney test, data not shown). Importantly, the phenotype was conserved across multiple differentiation lineages, as prematurely slowing MCM loading prior to endoderm differentiation also increased the number of cells positive for the endoderm transcription factor Sox17 relative to controls at the same time point (*Figure 7—figure supplement 1b*). To test if the pluripotency maintenance was due to Cdt1's role in origin licensing and not its mitotic or other functions (*Varma et al., 2012*), we slowed licensing by depleting the orthogonal MCM loading protein, Cdc6 (*Figure 7—figure supplement 3a–d*). A more modest Cdc6 knockdown correlated with a weaker, but detectable effect on MCM loading. Interestingly, this degree of licensing inhibition had no effect on G1 length. Despite the short G1 length, slowing MCM loading by depleting Cdc6 significantly promoted differentiation (*Figure 7—figure supplement 2b–f*, p<0.0001, two-tailed Mann-Whitney test. Thus, we conclude that slow MCM loading generally promoted differentiation and by extension, that rapid MCM loading preserves pluripotency.

## Discussion

In this study, we demonstrate that rapid MCM loading to license replication origins is an intrinsic property of pluripotent cells. Human embryonic stem cells have a remarkably fast MCM loading rate, and reprogramming to create induced pluripotent stem cells increases MCM loading rate. Moreover, MCM loading slows concurrently with the G1 lengthening and extensive cell cycle remodeling that accompany the early stages of differentiation (*Figure 7f*). To our knowledge, this is the first demonstration that the rate of MCM loading is developmentally regulated. Developmental regulation of MCM loading rate is consistent with previous work showing higher levels of total Cdt1 in asynchronous mouse ESCs than in differentiated cells (*Ballabeni et al., 2011*). The regulated decrease in MCM loading rate is critical during differentiation, as rapid MCM loading protects pluripotency, and prematurely slowing MCM loading promotes differentiation.

Pluripotent stem cells load MCM complexes rapidly to reach similar total amounts of loaded MCM at the G1/S transition in less time than their isogenic differentiated counterparts. Although we did not detect substantial MCM loading in telophase, as suggested previously (*Dimitrova et al.,*

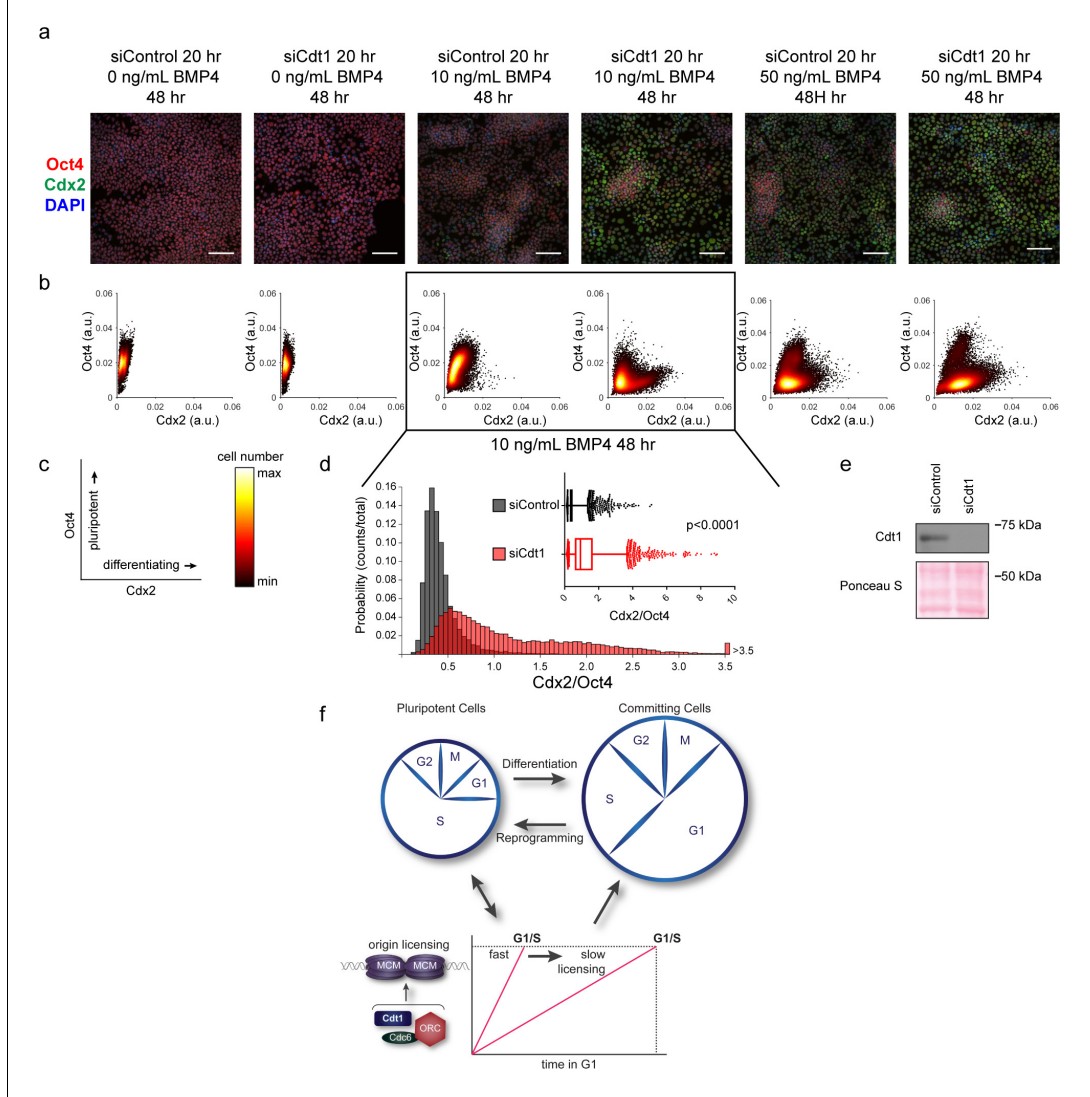

**Figure 7.** Slow MCM loading promotes differentiation. (**a**) Immunofluorescence microscopy of hESCs treated with 100 nM of siControl or 100 nM of siCdt1 for 20 hr and then treated with BMP4 as indicated. Cells were fixed and stained with DAPI (blue), Cdx2 antibody (green), and Oct4 antibody (red). Images are one region of 18 fields-of-view per condition; scale bar is 100 µm (see Materials and methods). (**b**) Density scatterplots of mean fluorescence intensity (arbitrary units) of Oct4 and Cdx2 staining for each cell in each condition, >18,000 cells were quantified per condition. See also *Figure 7—source data 1*. (**c**) Diagram of the relationship between Oct4 and Cdx2 in pluripotent and differentiated cells as plotted in (**b**); color bar for scatterplots in (**b**). (**d**) Histogram of mean fluorescence intensity ratio Cdx2/Oct4 for all cells in siControl and siCdt1 treated with 10 ng/mL BMP4 for 48 hr. Rightmost histogram bin contains all values greater than 3.5. The inset is a box-and-whiskers plot of the same data, center line is median, outer box edges are 25th and 75th percentile, whiskers edges are 1st and 99th percentile, individual data points are lowest and highest 1%, respectively. Medians are 0.3722, 0.9319, and means are 0.4285, 1.194 for siControl and siCdt1, respectively. Samples compared by two tailed Mann-Whitney test, ****p<0.0001. See also *Figure 7—source data 1*. (**e**) Immunoblot for Cdt1 in whole cell lysates at 20 hr of siRNA treatment, prior to BMP4 treatment. (**f**) Illustration of the relationship between differentiation and MCM loading rate changes.

DOI: https://doi.org/10.7554/eLife.30473.017

The following source data and figure supplements are available for figure 7:

**Source data 1.** Raw image values.

DOI: https://doi.org/10.7554/eLife.30473.021

**Figure supplement 1.** Complete microscopy dataset and endoderm differentiation.

DOI: https://doi.org/10.7554/eLife.30473.018

**Figure supplement 2.** Slow MCM loading promotes differentiation.

DOI: https://doi.org/10.7554/eLife.30473.019

**Figure supplement 3.** Reducing MCM loading rate by an alternative siRNA targeting Cdc6 instead of Cdt1.

*Figure 7 continued on next page*

*Figure 7 continued*

DOI: https://doi.org/10.7554/eLife.30473.020

*2002*), it is clear that telophase loading is an option in some cells and a requirement in cells with no detectable G1 such as *S. pombe* and in the first nuclear divisions in *D. melanogaster* (*Nishitani et al., 2000*; *Farrell and O'Farrell, 2014*). Stem cells achieve faster MCM loading, at least in part, by particularly high Cdt1 levels in G1. These high levels are achieved not only by a modest difference in *CDT1* mRNA (*Figure 3c*) but also post-transcriptionally by specific re-accumulation of Cdt1 protein in the preceding G2 phase (*Figure 5c*). Cdt1 stability in G2 phase has been attributed to geminin-mediated protection from the SCF$^{Skp2}$ E3 ubiquitin ligase in non-stem cells (*Tsunematsu et al., 2013*; *Ballabeni et al., 2004*; *Clijsters et al., 2013*). It is not clear, however, that geminin levels are particularly high in stem cells relative to differentiated cells (aside from differences expected from cell cycle distribution changes (*Figure 5a*) (*Ballabeni et al., 2011*)), so it seems unlikely that geminin drives higher Cdt1 levels in stem cells. Skp2 levels also do not change in the earliest stages of stem cell differentiation (*Egozi et al., 2007*). Cdt1 is protected in late S phase and G2 by cyclin A/Cdk1 activity (*Rizzardi et al., 2015*), and we thus consider it likely that the documented high CDK activity in stem cells contributes to Cdt1 stabilization in G2 (*Sela et al., 2012*). The anticipatory Cdt1 accumulation to promote MCM loading in G1 was originally proposed from experiments in cancer-derived cell lines (*Ballabeni et al., 2004*; *Clijsters et al., 2013*). Our observations in stem cells suggest this strategy is employed by non-transformed cells during developmental stages that require short G1.

Other factors besides Cdt1 accumulation may also accelerate MCM loading. The hESCs we assayed have 2–3 fold greater Cdt1 protein levels in G1 relative to NPCs yet load MCM 6.5 times faster per hour than NPCs. One Cdt1 molecule can (in vitro) load multiple MCM complexes since Cdt1 is released into the soluble phase immediately after completing a loading reaction (*Ticau et al., 2015*). Stem cells may experience less Cdc6 degradation in early G1 due to nearly constitutive Cyclin E/Cdk2 activity and/or attenuated APC$^{Cdh1}$ activity, corroborated by our observation that a Cdc6 mutant that is not targeted by APC$^{Cdh1}$ increases MCM loading rate in cells with long G1 phases (*Figure 6g*) (*Ballabeni et al., 2011*; *Neganova et al., 2009*; *Filipczyk et al., 2007*). Additionally, stem cells are enriched for euchromatin, an environment that may be particularly permissive for rapid MCM loading (*Chen and Dent, 2014*).

Rapid MCM loading may itself contribute to mechanisms that maintain short G1 phases in pluripotent cells. The origin licensing checkpoint links the amount of loaded MCM to G1 length by controlling Cdk2 activity. In that regard, overproducing Cyclin E 'short-circuited' the licensing checkpoint in slow loading differentiated cells. This checkpoint has thus far only been demonstrated in p53-proficient differentiated mammalian cells (*Ge and Blow, 2009*), but the G1 lengthening of hESCs after Cdt1 depletion suggests that pluripotent stem cells also have a functioning licensing checkpoint. Cells with fast MCM loading could satisfy this checkpoint quickly, activate Cyclin E/Cdk2, and thus spend less time in G1. Mechanisms that support short G1 length preserve pluripotency in hESCs (and promote reprogramming to iPSCs) since cells are most sensitive to differentiation cues in G1; in that regard, extending G1 phase in hESCs can increase differentiation propensity (*Soufi and Dalton, 2016*; *Pauklin and Vallier, 2013*; *Filipczyk et al., 2007*; *Coronado et al., 2013*). Recent work with quintuple knockout mice lacking all D and E type cyclins also reported that Cyclin E/Cdk2 further contributes to maintaining pluripotency by stabilizing the Oct4, Sox2, and Nanog transcription factors (*Liu et al., 2017*). Fast MCM loading may have evolved as an intrinsic property of pluripotent cells to maintain high Cdk2 activity and keep G1 phase short.

Cyclin/Cdk activity is not the sole connection between the cell cycle and pluripotency. Non-CDK cell-cycle-associated proteins regulate expression of key pluripotency genes including *SOX2* and *NANOG* (*Gonzales et al., 2015*; *Pauklin et al., 2016*; *Li et al., 2012*). Pluripotency transcription factors themselves regulate expression of cell cycle genes including those encoding cyclins, CDK inhibitors, and E2F3a (*Lee et al., 2010*; *Kanai et al., 2015*; *Choi et al., 2012*). On the other hand, pluripotency and cell cycle functions can be genetically uncoupled in experiments where manipulating the cell cycle did not alter pluripotency and vice versa (*Scognamiglio et al., 2016*; *Kareta et al., 2015b*). We observe that licensing inhibition can accelerate differentiation even without greatly

lengthening G1 (*Figure 7—figure supplements 2,3*) which may point to an additional direct and cell cycle-independent link between MCM loading rate and differentiation.

We note that early differentiation is not the only setting in which rapid MCM loading during a short G1 may be relevant. Like hESCs, activated T cells have very fast cell cycles with short G1 phases (*Kinjyo et al., 2015*). Oncogenic transformation is also frequently associated with G1 shortening. It may be that the pathways linking differentiation to MCM loading rate are also coopted in some cancers to induce rapid licensing. On the other hand, a subset of cancers may proliferate in a perpetually underlicensed state that contributes to the genome instability characteristic of transformed cells. Future investigations will elucidate the molecular relationships among developmental signaling pathways, MCM loading rate, and cell cycle remodeling.

# Materials and methods

**Key resources table**

| Reagent type (species) or resource | Designation | Source or reference | Identifiers | Additional information |
|---|---|---|---|---|
| Strain, strain background (*Escherichia coli*) | *E. coli*: DH5α | Invitrogen | Cat#11319009 | |
| Genetic reagent (*Homo sapiens*) | Rc/CMV cyclin E | *Hinds et al. (1992)* PMID: 1388095 | Addgene 8963 | |
| Genetic reagent (*Homo sapiens*) | pInducer20 | *Meerbrey et al., 2011* PMID: 21307310 | Addgene 44012 | |
| Genetic reagent (*Homo sapiens*) | ΔNRF | Dr. J. Bear | N/A | |
| Genetic reagent (*Homo sapiens*) | VSVG | Dr. J. Bear | N/A | |
| Genetic reagent (*Homo sapiens*) | pInducer20-Cyclin E1 | This Paper | N/A | see Materials and methods |
| Genetic reagent (*Homo sapiens*) | pDONR221 | Invitrogen | Cat#12536017 | |
| Genetic reagent (*Homo sapiens*) | pENTR221-Cyclin E1 | This Paper | N/A | |
| Genetic reagent (*Homo sapiens*) | PCR4-TOPO | Invitrogen | Cat# 450030 | |
| Genetic reagent (*Homo sapiens*) | pInducer20-blast | This Paper | N/A | see Materials and methods |
| Genetic reagent (*Homo sapiens*) | pInducer20-blast-Cdt1-HA | This Paper | N/A | see Materials and methods |
| Genetic reagent (*Homo sapiens*) | CLXSN-5myc-Cdc6-wt | This Paper | N/A | see Materials and methods |
| Genetic reagent (*Homo sapiens*) | CLXSN-5myc-Cdc6-mut | This Paper | N/A | see Materials and methods |
| Cell line (*Homo sapiens*) male | T98G | ATCC | Cat#CRL-1690 | |

*Continued on next page*

Continued

| Reagent type (species) or resource | Designation | Source or reference | Identifiers | Additional information |
|---|---|---|---|---|
| Cell line (*Homo sapiens*) female | RPE1-hTERT | ATCC | Cat#CRL-4000 | |
| Cell line (*Homo sapiens*) male | ARPE-19 | ATCC | Cat#CRL-2302 | |
| Cell line (*Homo sapiens*) female | H9 hESC (WA09) | WiCell | hPSCReg ID: WAe009-A | |
| Cell line (*Homo sapiens*) male | NPC | This Paper | N/A | see Materials and methods |
| Cell line (*Homo sapiens*) female | HEK293T | ATCC | Cat# CRL-3216 | |
| Cell line (*Homo sapiens*) male | ARPE-iPSC | This Paper | N/A | see Materials and methods |
| Antibody | Anti-Mcm2, mouse monoclonal (BM28) | BD Biosciences | Cat#610700;RRID: AB_2141952 | 1:10,000 (IB) 1:200 (FC) |
| Antibody | Anti-Mcm3, rabbit polyclonal | Bethyl Laboratories | Cat#A300-192A; RRID: AB_162726 | 1:10,000 (IB) 1:200 (FC) |
| Antibody | Anti-Cdt1, rabbit monoclonal (D10F11) (immunoblots) | Cell Signaling Technologies | Cat#8064S; RRID: AB_10896851 | 1:10,000 (IB) |
| Antibody | Anti-Cdt1, rabbit monoclonal (EPR17891) (flow cytometry) | Abcam | Cat#ab202067; RRID:AB_2651122 | 1:100 (FC) |
| Antibody | Anti-Cdc6, mouse monoclonal (180.2) | Santa Cruz Biotechnology | Cat#sc-9964; RRID: AB_627236 | 1:2000 (IB) |
| Antibody | Anti-Oct4, rabbit polyclonal (immunoblots) | Abcam | Cat#ab19857; RRID: AB_445175 | 1:4000 (IB) |
| Antibody | Anti-Oct4, mouse monoclonal (9B7) (microscopy) | Millipore | Cat#:MABD76; RRID: AB_10919170 | 1:1000 (IF) |
| Antibody | Anti-Cdx2, rabbit monoclonal (EPR2764Y) | Abcam | Cat#ab76541; RRID: AB_1523334 | 1:1000 (IF) |
| Antibody | Anti-Sox17, goat polyclonal | R and D Systems | Cat#AF1924; RRID: AB_355060 | 1:500 (IF) |
| Antibody | Anti-Cyclin E1, rabbit polyclonal | Santa Cruz Biotechnology | Cat#sc-198; RRID: AB_631346 | 1:2000 (IB) |
| Antibody | Anti-Orc1, rabbit polyclonal | Bethyl Laboratories | Cat#A301-892A; AB_1524103 | 1:1000 (IB) |
| Antibody | Anti-Orc6, rat monoclonal (3A4) | Santa Cruz Biotechnology | Cat#sc-32735; RRID: AB_670295 | 1:5000 (IB) |
| Antibody | Anti-geminin, rabbit polyclonal | Santa Cruz Biotechnology | Cat#sc-13015; RRID: AB_2263394 | 1:3000 (IB) |

*Continued*

| Reagent type (species) or resource | Designation | Source or reference | Identifiers | Additional information |
|---|---|---|---|---|
| Antibody | Anti-Histone H3, rabbit monoclonal (D1H2) | Cell Signaling Technologies | Cat#4499S; RRID: AB_10544537 | 1:10,000 (IB) |
| Antibody | Anti-TRA-1–60, mouse monoclonal (cl.A) | Invitrogen | Cat#41–1000; RRID: AB_605376 | 1:5000 (IB) |
| Antibody | Anti-nestin, mouse monoclonal (10 C2) | Abcam | Cat#ab22035; RRID: AB_446723 | 1:10000 (IB) |
| Antibody | Anti-TRA-1–60 mouse (immunofluorescence) | Millipore/Chemicon | Cat# MAB4360; RRID: AB_2119183 | 1:400 (IF) |
| Antibody | Anti-TRA-81 mouse (immunofluorescence) | Millipore/Chemicon | Cat# MAB4381; RRID:AB_177638 | 1:400 (IF) |
| Antibody | Anti-SSEA-4 mouse (MC-813–70) (immunofluorescence) | Millipore/Chemicon | Cat# MAB4304; RRID:AB_177629 | 1:200 (IF) |
| antibody | Anti-SSEA3 rabbit (MC-631) (immunofluorescence) | Millipore/Chemicon | Cat# MAB4303; RRID:AB_177628 | 1:200 (IF) |
| Antibody | Anti-Oct3/4 goat polyclonal (immunofluorescence) | Abcam | Cat# ab27985; RRID:AB_776898 | 1:200 (IF) |
| Antibody | Anti-NANOG goat polyclonal (immunofluorescence) | Everest Biotech | Cat# EB068601; RRID:AB_2150379 | 1:200 (IF) |
| Antibody | Anti-p27 rabbit polyclonal | Santa Cruz Biotechnology | Cat#sc-528; RRID:AB_632129 | 1:2000 (IB) |
| Antibody | Anti-α-tubulin | Sigma Aldrich | Cat#9026 | 1:50000 (IB) |
| Antibody | Goat anti-Mouse-HRP | Jackson ImmunoResearch | Cat#115-035-146; RRID: AB_2307392 | 1:10000 (IB) |
| Antibody | Donkey anti-Rabbit-HRP | Jackson ImmunoResearch | Cat#711-035-152; RRID: AB_10015282 | 1:10000 (IB) |
| Antibody | Bovine anti-Goat-HRP | Jackson ImmunoResearch | Cat#805-035-180; RRID: AB_2340874 | 1:10000 (IB) |
| Antibody | Donkey anti-Rat-HRP | Jackson ImmunoResearch | Cat#712-035-153; RRID: AB_2340639 | 1:10000 (IB) |
| Antibody | Donkey anti-Goat-Alexa 594 | Jackson ImmunoResearch | Cat#705-585-147; RRID: AB_2340433 | 1:1000 (IF) |
| Antibody | Donkey anti-Rabbit-Alexa 488 | Life Technologies | Cat#A21206; RRID: AB_2535792 | 1:1000 (IF) (FC) |
| Antibody | Goat anti-Mouse-Alexa 594 | Life Technologies | Cat#A11032; RRID: AB_2535792 | 1:1000 (IF) |
| Antibody | Donkey anti-Rabbit-Alexa 647 | Jackson ImmunoResearch | Cat#711-605-152; RRID: AB_2492288 | 1:1000 (FC) |
| Antibody | Donkey anti-Mouse-Alexa 488 | Jackson ImmunoResearch | Cat#715-545-150; RRID: AB_2340845 | 1:1000 (FC) |
| Sequence-based reagent | siCdt1- CCUACGUCAA GCUGGACAATT | *Nevis et al. (2009)* PMCID: PMC2972510 | N/A | |
| Sequence-based reagent | siCdc6-2534- CACCAUGCUCAGCC AUUAAGGUAUU | *Nevis et al. (2009)* PMCID: PMC2972510 | N/A | |

Continued

| Reagent type (species) or resource | Designation | Source or reference | Identifiers | Additional information |
|---|---|---|---|---|
| Sequence-based reagent | siCdc6-2144- UCUAGCCAAUGUGC UUGCAAGUGUA | *Nevis et al. (2009)* PMCID: PMC2972510 | N/A | |
| Sequence-based reagent | siControl (Luciferase)- CUUACGCUGA GUACUUCGA | *Coleman et al. (2015)* PMID: 26272819 | N/A | |
| Sequence-based reagent | siMCM3-2859 5'- augacuauu gcaucuucauug | This paper | | synthesized by invitrogen |
| Sequence-based reagent | siMCM3-2936 5'- aacauauga cuucugaguacu | This paper | | synthesized by invitrogen |
| Sequence-based reagent | POU5F1-F: 5'-CCTGAAGCAGA AGAGGATCACC, | Eton Bioscience | | |
| Sequence-based reagent | POU5F1-R 5'-AAAGCGGCAGA TGGTCGTTTGG, | Eton Bioscience | | |
| Sequence-based reagent | CDX2-F 5'-ACAGTCGCTA CATCACCATCCG, | Eton Bioscience | | |
| Sequence-based reagent | CDX2-R 5'-CCTCTCCT TTGCTCTGCGGTTC, | Eton Bioscience | | |
| Sequence-based reagent | T-F 5'-CTTCAGCA AAGTCAAGCTCACC, | Eton Bioscience | | |
| Sequence-based reagent | T-R 5'-TGAACTGGGTCT CAGGGAAGCA, | Eton Bioscience | | |
| Sequence-based reagent | SOX17-F 5'-ACGCTTTCA TGGTGTGGGCTAAG, | Eton Bioscience | | |
| Sequence-based reagent | SOX17-R 5'-GTCAGCGC CTTCCACGACTTG, | Eton Bioscience | | |
| Sequence-based reagent | CDT1-F 5'-GGAGGTCAGAT TACCAGCTCAC, | Eton Bioscience | | |
| Sequence-based reagent | CDT1-R, 5'-TTGACGTGC TCCACCAGCTTCT, | Eton Bioscience | | |
| Sequence-based reagent | SOX2-F 5'-CTACAGCAT GATGCAGGACCA, | Eton Bioscience | | |
| Sequence-based reagent | SOX2-R 5'-TCTGCGAGCT GGTCATGGAGT, | Eton Bioscience | | |
| Sequence-based reagent | PAX6-F 5'-AATCAGAG AAGACAGGCCA, | Eton Bioscience | | |
| Sequence-based reagent | PAX6-R 5'-GTGTAGGTA TCATAACTC, | Eton Bioscience | | |
| Sequence-based reagent | ACTB-F 5'-CACCATTGGC AATGAGCGGTTC, | Eton Bioscience | | |
| Sequence-based reagent | ACTB-R 5'-AGGTCTTTGC GGATGTCCACGT | Eton Bioscience | | |
| Sequence-based reagent | CDC6-KEN-F: 5- ctccaccaaagc aaggcaaggcggc cgcaggtccccc tcactcacatacac | Eurofins | | |
| Sequence-based reagent | CDC6-KEN-R: 5- GTGTATGTGAGTGAGG GGGACCTGCG GCCGCCTTGCCTTGCTTTGGTGGAG | Eurofins | | |
| Sequence-based reagent | CDC6-DBOX-F: 5- aagccctgcctct cagccccgccaaacgt gccggcgatgacaa cctatgcaa | Eurofins | | |

*Continued*

| Reagent type (species) or resource | Designation | Source or reference | Identifiers | Additional information |
|---|---|---|---|---|
| Sequence-based reagent | CDC6-DBOX-R: 5- TTGCATAGGTTGTCA TCGCCGGCACGTTT GGCGGGGCTGA GAGGCAGGGCTT | Eurofins | | |
| Sequence-based reagent | AgeI-rta3-F: 5- gctcggatctccacc ccgtaccggtcctg cagtcgaattcac | Eurofins | | |
| Sequence-based reagent | AgeI-IRES-blast-R: 5- ACAAAGGCTTGGC CATGGTTTAAGCTTATCA TCGTGTTTTTCA | Eurofins | | |
| Sequence-based reagent | Blast-F:5- tgaAaaacacgat gataagctt aaaccatggc caagcctttgt | Eurofins | | |
| Sequence-based reagent | Blast-AgeI-Ind-R: 5- GTTCAATCATGG TGGACCGG CTATTAGCCCTCCCAC ACATAACCA | Eurofins | | |
| Sequence-based reagent | BP-cycE-F 5' GGGGACAAGTTTGTAC AAAAAAGCAGGC TACCATGAAGGAG GACGGCGGC | Eurofins | | |
| Sequence-based reagent | BP-cycE-R 5' GGGGACCACTTTG TACAAGAAAGCTGG GTTCACGCCATT TCCGGCCCGCT | Eurofins | | |
| Software, algorithm | MATLAB | MathWorks | https://www.math works.com/ | |
| Software, algorithm | GraphPad Prism 7 | GraphPad Software | https://www.graphpad .com/scientific- software/prism/ | |
| Software, algorithm | NIS-Elements Advanced Research Software | Nikon | https://www.nikonin struments .com/Products /Software/ NIS-Elements- Advanced-Research | |
| Software, algorithm | CellProfiler | *Carpenter et al., 2006* PMC1794559 | http://cellprofiler.org/ | |
| Software, algorithm | FCS Express 6 | De Novo Software | https://www.denovo software.com/ | |
| Software, algorithm | FCSExtract Utility | Earl F Glynn | http://research. stowers.org/mcm /efg/Scientific Software/ Utility/FCSExtract/ index.htm | |
| Software, algorithm | QUMA | RIKEN | http://quma. cdb. riken.jp | |
| Software, algorithm | Adobe Photoshop CS6 | Adobe | http://www.adobe. com/ products/ photoshop.html | |

*Continued on next page*

Continued

| Reagent type (species) or resource | Designation | Source or reference | Identifiers | Additional information |
|---|---|---|---|---|
| Commercial assay or kit | CytoTune-iPS 2.0 Sendai reprogramming kit | Invitrogen | Cat#A16517 | |
| Commercial assay or kit | DNeasy Blood and Tissue kit | Qiagen | Cat#69504 | |
| Commercial assay or kit | RNeasy Mini kit | Qiagen | Cat#74104 | |
| Commercial assay or kit | Epitect Bisulfite kit | Qiagen | Cat#59104 | |
| Commercial assay or kit | Norgen Biotek's Total RNA Purification Kit | Norgen Biotek | Cat#37500 | |
| Commercial assay or kit | Applied Biosystem's High-Capacity RNA-to-cDNA | Applied Biosystem | Cat#4387406 | |
| Commercial assay or kit | Alkaline Phosphatase Detection Kit | Millipore | Cat# SCR004 | |
| Commercial assay or kit | QIAquick Gel Extraction kit | Qiagen | Cat# 28704 | |
| Chemical compound, drug | DAPI | Life Technologies | Cat#D1306 | |
| Chemical compound, drug | EdU | Santa Cruz Biotechnology | Cat#sc-284628 | |
| Chemical compound, drug | Ponceau S | Sigma Aldrich | Cat#P7170-1L | |
| Peptide, recombinant protein | BMP4 Protein | R and D Systems | Cat#314 BP-010 | |
| Peptide, recombinant protein | Activin A Protein | R and D Systems | Cat#338-AC-010 | |
| Chemical compound, drug | Y-27632 2HCl | Selleck Chemicals | Cat#S1049 | |
| Chemical compound, drug | CHIR-99021 | Selleck Chemicals | Cat#S2924 | |
| Chemical compound, drug | mTESR1 | Stem Cell Technologies | Cat#05850 | |
| Chemical compound, drug | STEMdiff Neural Induction Medium | Stem Cell Technologies | Cat#05835 | |
| Chemical compound, drug | STEMdiff Neural Progenitor Medium | Stem Cell Technologies | Cat#05833 | |
| Chemical compound, drug | Essential 8 Medium | Life Technologies | Cat#A1517001 | |
| Chemical compound, drug | Doxycycline | CalBiochem | Cat#324385 | |
| Chemical compound, drug | Alexa 647-azide | Life Technologies | Cat#A10277 | |
| Chemical compound, drug | Alexa 488-azide | Life Technologies | Cat#A10266 | |
| Chemical compound, drug | Hydroxyurea | Alfa Aesar | Cat#A10831 | |
| Chemical compound, drug | Corning Matrigel GFR Membrane Matrix | Corning | Cat#CB-40230 | |
| Chemical compound, drug | Poly-L-Ornithine | Sigma Aldrich | Cat#P4957-50ML | |

*Continued*

| Reagent type (species) or resource | Designation | Source or reference | Identifiers | Additional information |
|---|---|---|---|---|
| Chemical compound, drug | Laminin | Sigma Aldrich | Cat#L2020-1MG | |
| Chemical compound, drug | ReLesR | Stem Cell Technologies | Cat#05872 | |

## Cell culture

Cell lines were authenticated by STR profiling (ATCC, Manassas, VA) and confirmed to be myco-plasma negative. T98G, HEK293T, and RPE1-hTERT were cultured in Dulbecco's Modified Eagle Medium (DMEM) supplemented with 2 mM L-glutamine and 10% fetal bovine serum (FBS) and incubated in 5% CO2 at 37°C. ARPE-19 (male) were cultured in 1:1 DMEM:F12 supplemented with 2 mM L-glutamine and 10% fetal bovine serum and incubated in 5% CO2 at 37°C. T98G, HEK293T, RPE1-hTERT, and ARPE-19 cells were from the ATCC and were passaged with trypsin and not allowed to reach confluency. WA09 (H9 hESCs) were cultured in mTeSR1 (StemCell Technologies) with media changes every 24 hr on Matrigel (Corning, New York, NY) coated dishes and incubated in 5% CO2 at 37°C. H9s had normal diploid karyotype at passage 32 and were used from passage 32–42. ARPE-iPSCs were cultured in Essential 8 (Life Technologies, Grand Island, NY) with media changes every 24 hr on Matrigel (Corning) coated dishes and incubated in 5% CO2 at 37°C. iPSCs were used from passage 20–25 and had normal karyotype. Both hESCs and iPSCs were routinely passaged every 4 days as aggregates using ReLeSR, according to manufacturer's instructions (StemCell Technologies, Canada). The hESCs and iPSCs were only passaged as single cells in 10 μM Y-27632 2HCl (Selleck Chemicals, Houston, TX) for experiments, as described previously (*Watanabe et al., 2007*). NPCs were cultured in Neural Progenitor Medium (StemCell Technologies) with media changes every 24 hr on poly-L-ornithine/Laminin (Sigma Aldrich, St. Louis, MO) coated dishes and incubated in 5% $CO_2$ at 37°C. NPCs were passaged with StemPro Accutase (Gibco, Waltham, MA) weekly.

## Total lysate and chromatin fractionation

Cells were collected via trypsinization. For total protein lysates, cells were lysed on ice for 20 min in CSK buffer (300 mM sucrose, 100 mM NaCl, 3 mM $MgCl_2$, 10 mM PIPES pH 7.0) with 0.5% triton x-100 and protease and phosphatase inhibitors (0.1 mM AEBSF, 1 μg/ mL pepstatin A, 1 μg/ mL leupeptin, 1 μg/ mL aprotinin, 10 μg/ ml phosvitin, 1 mM β-glycerol phosphate, 1 mM Na- orthovanadate). Cells were centrifuged at 13,000 xg at 4°C for 5 min, then the supernatants were transferred to a new tube for a Bradford Assay (Biorad, Hercules, CA) using a BSA standard curve. Chromatin fractionation for immunoblotting was performed as described previously (*Cook et al., 2002*; *Méndez and Stillman, 2000*), using CSK buffer with 1 mM ATP, 5 mM $CaCl_2$, 0.5% triton x-100 and protease and phosphatase inhibitors to isolate insoluble proteins and S7 nuclease (Roche) to release DNA bound proteins. A Bradford Assay (Biorad) was performed for equal loading. For 100 mM or 300 mM NaCl soluble/pellet fractionation, cells were lysed in standard CSK (100 mM NaCl) or high-salt CSK (300 mM NaCl) with 0.5% triton X-100 with protease and phosphatase inhibitors for 5 min on ice. Then cells were centrifuged at 2000 xg for 3 min, supernatants transferred to a new tube as the soluble fraction. The remaining pellet was suspended in 2x SDS loading buffer (2% SDS, 5% 2-mercaptoethanol, 0.1% bromophenol blue, 50 mM Tris pH 6.8, 10% glycerol) as the pellet fraction. Bradford assay was performed on the soluble fraction for equal loading.

## Immunoblotting

Samples were diluted with SDS loading buffer (final: 1% SDS, 2.5% 2-mercaptoethanol, 0.1% bromophenol blue, 50 mM Tris pH 6.8, 10% glycerol) and boiled. Samples were run on SDS-PAGE gels, then the proteins transferred onto polyvinylidene difluoride membranes (Thermo Fisher, Waltham, MA) or nitrocellulose (GE Healthcare, Chicago, IL). Membranes were blocked at room temperature for 1 hr in either 5% milk or 5% BSA in Tris-Buffered-Saline-0.1%-tween-20 (TBST). After blocking, membranes were incubated in primary antibody overnight at 4°C in either 1.25% milk or 5% BSA in TBST with 0.01% sodium azide. Blots were washed with TBST then

incubated in HRP-conjugated secondary antibody in either 2.5% milk or 5% BSA in TBST for 1 hr, washed with TBST, and then membranes were incubated with ECL Prime (Amersham, United Kingdom) and exposed to autoradiography film (Denville, Holliston, MA). Equal protein loading was verified by Ponceau S staining (Sigma Aldrich). Antibodies used for immunoblotting were: Mcm2, (BD Biosciences, San Jose, CA, Cat#610700), Mcm3, (Bethyl Laboratories, Montgomery, TX, Cat#A300-192A), Cdt1, (Cell Signaling Technologies, Beverly, MA, Cat#8064S), Cdc6, (Santa Cruz Biotechnology, Santa Cruz, CA, Cat#sc-9964), Oct4, (Abcam, Cambridge, MA, Cat#ab19857),Cyclin E1, (Santa Cruz Biotechnology, Cat#sc-198), Orc1, (Bethyl Laboratories, Cat#A301-892A), Orc6, (Santa Cruz Biotechnology, Cat#sc-32735), geminin, (Santa Cruz Biotechnology, Cat#sc-13015), Histone H3, (Cell Signaling Technologies, Cat#4499S), TRA-1–60, (Invitrogen, Cat#41–1000), nestin, (Abcam, Cat#ab22035), p27 (Santa Cruz Biotechnology, Cat#sc-528), α-tubulin (Sigma Aldrich, Cat#9026).

## Flow cytometry

For EdU-labeled samples, cells were incubated with 10 uM EdU (Santa Cruz Biotechnology) for 30 min prior to collection. For total protein flow cytometry, cells were collected with trypsin and resuspended as single cells, washed with PBS, and fixed with 4% paraformaldehyde (Electron Microscopy Sciences, Hatfield, PA) in PBS for 15 min at room temperature, then 1% BSA-PBS was added, mixed and cells were centrifuged at 1000 xg for 7 min (and for all following centrifuge steps) then washed with 1% BSA-PBS and centrifuged. Fixed cells were permeabalized with 0.5% triton x-100 in 1% BSA-PBS at room temperature for 15 min, centrifuged, then washed once with 1% BSA, PBS and centrifuged again before labeling. For chromatin flow cytometry, cells were collected with trypsin and resuspended as single cells, washed with PBS, and then lysed on ice for 5 min in CSK buffer with 0.5% triton x-100 with protease and phosphatase inhibitors. Next, 1% BSA-PBS was added and mixed, then cells were centrifuged for 3 min at 1000 xg, then fixed in 4% paraformaldehyde in PBS for 15 min at room temperature. For 100 mM NaCl vs 300 mM NaCl CSK, cells were processed as in chromatin flow cytometry above, except CSK containing 300 mM NaCl was used instead of the normal 100 mM NaCl. Next, 1% BSA-PBS was added, mixed and cells were centrifuged then washed again before labeling. The labeling methods for total protein samples and chromatin samples were identical. For DNA synthesis (EdU), samples were centrifuged and incubated in PBS with 1 mM CuSO$_4$, 1 µM fluorophore-azide, and 100 mM ascorbic acid (fresh) for 30 min at room temperature in the dark. 1% BSA-PBS +0.1% NP-40 was added, mixed and centrifuged. Samples were resuspended in primary antibody in 1% BSA-PBS +0.1% NP-40 and incubated at 37°C for 1 hr in the dark. Next, 1% BSA-PBS +0.1% NP-40 was added, mixed and centrifuged. Samples were resuspended in secondary antibody in 1% BSA-PBS +0.1% NP-40 and incubated at 37°C for 1 hr in the dark. Next, 1% BSA-PBS +0.1% NP-40 was added, mixed and centrifuged. Finally, cells were resuspended in 1% BSA-PBS +0.1% NP-40 with 1 µg/mL DAPI (Life Technologies) and 100 µg/mL RNAse A (Sigma Aldrich) and incubated overnight at 4°C in the dark. Samples were run on a CyAn ADP flow cytometer (Beckman Coulter, Brea, CA) and analyzed with FCS Express six software (De Novo, Glendale, CA). The following antibody/fluorophore combinations were used: (1): Alexa 647-azide (Life Technologies), primary: Mcm2 (BD Biosciences, Cat#610700), secondary: Donkey anti-mouse-Alexa 488 (Jackson ImmunoResearch), DAPI. (2): Alexa 488-azide (Life Technologies), primary: Cdt1 (Abcam, Cat#610700), secondary: Donkey anti-rabbit-Alexa 647 (Jackson ImmunoResearch), DAPI. (3): Alexa 647-azide (Life Technologies), primary: Mcm3 (Bethyl Cat#A300-192A), secondary: Donkey anti-rabbit-Alexa 488 (Jackson ImmunoResearch), DAPI. (4): primary: Mcm3 (Bethyl Cat#A300-192A), Mcm2 (BD Biosciences, Cat#610700), secondary: Donkey anti-mouse-Alexa 488, Donkey anti-rabbit-Alexa 647 (Jackson ImmunoResearch, West Grove, PA), DAPI. Cells were gated on FS-area vs SS-area. Singlets were gated on DAPI area vs DAPI height. The positive/negative gates for EdU and MCM were gated on a negative control sample, which was treated with neither EdU nor primary antibody, but incubated with 647-azide and the secondary antibody Donkey anti-mouse-Alex 488 and DAPI to account for background staining (*Figure 1—figure supplement 1*).

## Doubling time

Doubling time was calculated by plating equal number of cells as described above and counting cell number over time using a Luna II automated cell counter (Logos Biosystems, South Korea) at 24, 48, and 72 hr after plating. Three or four wells were counted as technical replicates at each timepoint. GraphPad Prism's regression analysis was used to compute doubling time, and multiple biological replicates were averaged for a final mean doubling time. ARPE-19s were counted four times, hESCs and NPCs three times, and iPSCs two times.

## Cell synchronization and treatments

To synchronize cells in G1, T98G cells were grown to 100% confluency, washed with PBS, and incubated for 72 hr in 0.1% FBS, DMEM, L-glutamine. After serum-starvation, cells were re-stimulated by passaging 1:3 with trypsin to new dishes in 20% FBS, DMEM, L-glutamine, collecting cells 10 hr and 12 hr post-stimulation. To synchronize T98G cells in early S phase, cells were treated as for G1, except 1 mM Hydroxyurea (Alfa Aesar, Haverhill, MA) was added to the media upon re-stimulating and cells were collected 18 hr post-stimulation. To synchronize cells in mid-late S, cells were treated as in early S, then at 18 hr post-stimulation cells were washed with PBS and released into 10% FBS, DMEM, L-Glutamine, collecting 6 hr, 8 hr post release.

To synchronize RPE1-hTERT cells in G0, cells were grown to 100% confluency, then incubated for 48 hr in 10% FBS, DMEM, L-glutamine. For RPE1 in G1/S, G0 cells were trypsinized and passaged 1:6 with trypsin to new dishes in 10% FBS, DMEM, L-glutamine, and collected with trypsin 22 hr later. For cycloheximide (Sigma) treatment, asynchronous RPE cells were treated with 10 ug/mL for 4 hr or 8 hr as indicated. For UV irradiation, asynchronous RPE cells were treated with 20 J/m$^2$ of UV with a Stratalinker (Stratagene, San Diego, CA) and collected 1 hr later.

## Cloning

The pInducer20-Cyclin E plasmid was constructed using the Gateway cloning method (Invitrogen). The attB sites were added to Cyclin E1 cDNA by PCR using Rc/CMV cyclin E plasmid as a template and BP-cycE-F (5' GGGGACAAGTTTGTACAAAAAAGCAGGCTACCATGAAGGAGGACGGCGGC) and BP-cycE-R primers (5' GGGGACCACTTTGTACAAGAAAGCTGGGTTCACGCCA TTTCCGGCCCGCT) (Hinds et al., 1992). The PCR product was recombined with pDONR221 plasmid using BP clonase (Invitrogen) according to the manufacturer's instructions and transformed into DH5α to create pENTR221-Cyclin E1. Then the LR reaction was performed between pInducer20 and pENTR221-Cyclin E1 using LR Clonase (Invitrogen, Carlsbad, CA) according to manufacturer's instructions and transformed into DH5α to create pInducer20-Cyclin E1. The pInducer20 plasmid was converted to blasticidin resistance (pInducer20-blast2) by Gibson Assembly (New England Biolabs, Ipswitch, MA) following manufacturer's protocol. pInducer20 was cut with AgeI and assembled with PCR products with the following primers:

AgeI-rta3-F: 5- gctcggatctccacccgtaccggtcctgcagtcgaattcac
AgeI-IRES-blast-R: 5- ACAAAGGCTTGGCCATGGTT TAAGCTTATCATCGTGTTTTTCA
Blast-F:5- tgaAaaacacgatgataagcttaaaccatggccaagcctttgt
Blast-AgeI-Ind-R: 5- GTTCAATCATGGTGGACCGG CTATTAGCCCTCCCACACATAACCA

The pLenti CMV blast plasmid was a template for the blasticidin resistance gene. A tagged Cdt1-HA was cloned into pInducer20-blast using Gateway cloning as described above.

The Cdc6 mutant unable to bind APC$^{CDH1}$ (5myc-Cdc6-mut) was described previously: R56A, L59A, K81A, E82A, N83A (Petersen et al., 2000). pCLXSN-5myc-Cdc6-wt was cloned to 5myc-Cdc6-mut by two sequential Gibson assemblies (NEB) according to manufacturer's instructions. Primers used:

CDC6-KEN-F: 5- ctccaccaaagcaaggcaaggcggccgcaggtcccctcactcacatacac
CDC6-KEN-R: 5- GTGTATGTGAGTGAGGGGGACCTGCGGCCGCCTTGCCTTGCTTTGGTGGAG
CDC6-DBOX-F: 5- aagccctgcctctcagcccgccaaacgtgccggcgatgacaacctatgcaa
CDC6-DBOX-R: 5- TTGCATAGGTTGTCATCGCCGGCACGTTTGGCGGGGCTGAGAGG-CAGGGCTT

## Cell line construction and inducible protein production

To package retrovirus, pCLXSN 5myc-Cdc6 wt or mut were co-transfected with pCI-GPZ and VSVG plasmids into HEK293T using 50 µg/mL Polyethylenimine-Max (Aldrich Chemistry).To package lentivirus, pInducer20-Cyclin E1 or pInducer20-blast2-Cdt1-HA wwere co-transfected with ΔNRF and VSVG plasmids into HEK293T using 50 µg/mL Polyethylenimine-Max (Aldrich Chemistry). Viral supernatant was transduced with 8 ug/mL Polybrene (Millipore, Burlington, MA) onto RPE1-hTERT cells overnight. Cells were selected with 500 ug/mL neomycin (Gibco) or 5 µg/mL blasticidin (Research Products International, Mount Prospect, IL) for 1 week. To overproduce Cyclin E1, cells were treated with 100 ng/mL doxycycline (CalBiochem, San Diego, CA) for 72 hr in 10% FBS, DMEM, L-glutamine. Control cells were the Inducer20-Cyclin E1 without doxycycline. To overproduce Cdt1, cells were treated with 100 ng/mL doxycycline for 48 hr in 10% FBS, DMEM, L-glutamine, control cells were without doxycycline.

## siRNA transfections

For siRNA treatment, Dharmafect 1 (Dharmacon, Lafayette, CO) was mixed in mTeSR1 with the appropriate siRNA according to the manufacturer's instructions, then diluted with mTeSR1 and added to cells after aspirating old media. The final siRNA concentrations were: 100 nM siControl (Luciferase), 25 or 100 nM siCdt1, or a mixture of two siCdc6 (2144 and 2534 at 50 nM each). The Cdt1 siRNA mix was incubated on cells for either 20 or 24 hr, then changed to new mTeSR1 without siRNA. The Cdc6 siRNA mix was incubated on cells for 24 hr, then changed to new mTeSR1 without siRNA for 8 hr (32 total hours). The Cdt1, Cdc6 and Luciferase siRNA were described previously (*Coleman et al., 2015*; *Nevis et al., 2009*). For siRNA treatment of RPE cells, Dharmafect 1 (Dharmacon) was mixed in Optimem (Gibco) with the appropriate siRNA according to manufacturer's instructions, then diluted with DMEM, 10% FBS, L-glutamine and added to cells after aspirating old media. The next day, the siRNA mix was aspirated and replaced with fresh DMEM, 10% FBS, L-glutamine, collecting samples 72 hr after the start of siRNA treatment. The siRNA were siControl (Luciferase) at 100 nM or a mixture of two MCM3 siRNA (2859 and 2936 at 50 nM each).

siMCM3-2859 5'- augacuauugcaucuucauugdTdT
siMCM3-2936 5'- aacauaugacuucugaguacudTdT

## Differentiation

Mesoderm (BMP4): hESCs were passaged as single cells at $7 \times 10^3$/ cm$^2$ in mTeSR1 with 10 µM Y-27632 2HCl onto Matrigel-coated plates. 24 hr later, the media was changed to start differentiation with fresh mTeSR1 with 100 ng/mL BMP4 (R and D Systems, Minneapolis, MN), and 24 hr later the media was changed to fresh mTeSR1 with 100 ng/mL BMP4 for 48 total hours of differentiation.

Neuroectoderm: hESCs were differentiated using a monolayer-based protocol in Neural Induction Medium: hESCs were passaged as single cells at $5.2 \times 10^4$/ cm$^2$ in STEMdiff Neural Induction Medium (StemCell Technologies) with 10 µM Y-27632 2HCl onto Matrigel coated plates, and plating started the differentiation. 24 hr later, the media was changed to fresh Neural Induction Medium for another 24 hr for 48 hr total differentiation. To derive NPCs, hESCs were differentiated in Neural Induction medium (StemCell Technologies) using the Embryoid Body Neural Induction protocol according to manufacturer's instructions, similar to previous reports (*Robinson et al., 2016*). Once generated, NPCs were maintained in Neural Progenitor Medium (StemCell Technologies).

Mesoderm (GSK3βi): hESCs were passaged as single cells at $3 \times 10^4$/ cm$^2$ in mTeSR1 with 10 µM Y-27632 2HCl onto Matrigel-coated plates. 24 hr later, the media was changed to start differentiation. Cells were washed with Advanced RPMI 1640 (Gibco), then incubated in Advanced RPMI 1640 with B27 minus insulin (Gibco), 2 mM L-glutamine, and 8 µM CHIR-99021 (Selleck Chemicals). At 24 hr after changing the media, cells were washed with Advanced RPMI 1640 then incubated in Advanced RPMI 1640 with B27 minus insulin (Gibco), 2 mM L-glutamine, without CHIR-99021 for 24 hr for a total of 48 hr of differentiation.

Endoderm: hESCs were passaged as single cells at $4 \times 10^3$/ cm$^2$ in mTeSR1 with 10 µM Y-27632 2HCl onto Matrigel-coated plates. The next day, the media was changed to fresh mTeSR1 without Y-27632 2HCl. 24 hr later, the media was changed to start differentiation. The cells were washed with Advanced RPMI 1640 (Gibco), then incubated in Advanced RPMI 1640 with 0.2% FBS, 2 mM L-glutamine, 100 ng/mL Activin A (R and D Systems) and 2.5 µM CHIR-99021. At 24 hr after

changing the media, cells were washed with Advanced RPMI 1640 then incubated in Advanced RPMI 1640 with 0.2% FBS, 2 mM L-glutamine, 100 ng/mL Activin A, without CHIR-99021 for 24 hr for a total of 48 hr of differentiation.

## Phase contrast microscopy

Phase contrast images were acquired with an Axiovert 40 CFL inverted microscope, 20x objective (Zeiss, Germany).

## Immunofluorescence microscopy

For immunofluorescence microscopy, hESCs were plated as single cells in mTeSR1 with 10 µM Y-27632 2HCl in Matrigel-coated, 24 well, #1.5 glass bottom plates (Cellvis) at $7 \times 10^3/$ cm$^2$ for siCdt1, Mesoderm (BMP4), at $5 \times 10^3$ for siCdc6, Mesoderm (BMP4), and at $4 \times 10^3/$ cm$^2$ for siCdt1, Endoderm. Cells were incubated with siCdt1 for 20 hr (Mesoderm (BMP4)), 24 hr (Endoderm) or siCdc6 for 32 hr (Mesoderm [BMP4]) all in parallel with siControl as described above (siRNA transfections). After siRNA treatment, cells were differentiated as described above (Differentiation) with the following modifications: For Mesoderm (BMP4), multiple BMP4 concentrations and treatment times were used as indicated (*Figure 7*, *Figure 7—figure supplement 1*). For treatment less than 48 hr, cells were incubated in mTeSR1 after siRNA treatment until starting differentiation. (Example: 12 hr of mTeSR1 then 36 hr of BMP4, for a total of 48 hr). For endoderm, the first RPMI/Activin/CHIR-99021 was immediately after siRNA, without a day of incubation in mTeSR1. After differentiation, cells were fixed in 4% paraformaldehyde in PBS for 15 min at room temperature, washed with PBS, and permeabalized with 5% BSA, PBS, 0.3% triton x-100 at 4°C overnight. Next, cells were incubated in primary antibody in 5% BSA, PBS, 0.3% triton x-100 at 4°C overnight. Cells were washed with PBS at room temperature, then incubated in secondary antibody in 5% BSA, PBS, 0.3% triton x-100 at room temperature for 1 hr. Cells were washed with PBS, then incubated in 1 µg/mL DAPI in PBS for 10 min at room temperature, then washed with PBS. For Mesoderm (BMP4)the primary antibodies were Oct4 (Millipore, Cat#MABD76) and Cdx2 rabbit (Abcam, Cat#ab76541), the secondary antibodies were goat anti-mouse-Alexa 594, donkey anti-rabbit-Alexa 488. For endoderm the primary antibody was Sox17 (R and D Systems, Cat#AF1924), the secondary antibody was donkey anti-goat-Alexa 594 (Jackson ImmunoResearch). Cells were imaged in PBS on a Nikon Ti Eclipse inverted microscope with an Andor Zyla 4.2 sCMOS detector. Images were taken as $3 \times 3$ scan of 20x fields with a 0.75 NA objective, stitched with 15% overlap between fields using NIS-Elements Advanced Research Software (Nikon, Japan). Shading correction was applied within the NIS-Elements software before acquiring images. Raw images were quantified using a custom CellProfiler pipeline.

## qPCR (*Figure 3*)

RNA lysates were prepared using Norgen Biotek's Total RNA Purification Kit (Cat. 37500). Lysates were first treated with Promega RQ1 RNase-Free DNase (Promega, Madison, WI), and then converted to cDNA using Applied Biosystem's High-Capacity RNA-to-cDNA Kit (Cat. 4387406). Quantitative real-time PCR (qPCR) with SYBR Green (Bio-Rad; SsoAdvanced Universal SYBR Green Supermix, Cat. 1725271) was carried out to assess gene expression. All results were normalized to *ACTB*. Primers for qPCR were ordered from Eton Bioscience, San Diego, CA. Primers:

POU5F1-F: 5'-CCTGAAGCAGAAGAGGATCACC,
POU5F1-R 5'-AAAGCGGCAGATGGTCGTTTGG,
CDX2-F 5'-ACAGTCGCTACATCACCATCCG,
CDX2-R 5'-CCTCTCCTTTGCTCTGCGGTTC,
T-F 5'-CTTCAGCAAAGTCAAGCTCACC,
T-R 5'-TGAACTGGGTCTCAGGGAAGCA,
SOX17-F 5'-ACGCTTTCATGGTGTGGGCTAAG,
SOX17-R 5'-GTCAGCGCCTTCCACGACTTG,
CDT1-F 5'-GGAGGTCAGATTACCAGCTCAC,
CDT1-R, 5'-TTGACGTGCTCCACCAGCTTCT,
SOX2-F 5'-CTACAGCATGATGCAGGACCA,
SOX2 -R 5'-TCTGCGAGCTGGTCATGGAGT,
PAX6-F 5'-AATCAGAGAAGACAGGCCA,

PAX6-R 5'-GTGTAGGTATCATAACTC,
ACTB-F 5'-CACCATTGGCAATGAGCGGTTC,
ACTB-R 5'-AGGTCTTTGCGGATGTCCACGT.

## Generating ARPE-19-iPS cells

ARPE-19-iPS cells were derived from the human retinal pigment epithelial cell line ARPE-19 by reprogramming with CytoTune-iPS 2.0 Sendai reprogramming kit (Invitrogen) following the manufacturer's instructions.

Briefly, two days before Sendai virus transduction, 100,000 ARPE-19 cells were plated into one well of a 6-well plate with ATCC-formulated DMEM:F12 medium and were transduced with the CytoTune 2.0 Sendai reprogramming vectors at the MOI recommended by the manufacturer 48 hr later (d0). The medium was replaced with fresh medium every other day starting from one day after transduction (d1). At day 7, transduced cells were replated on Matrigel-coated six-well plates. Cells were fed with Essential eight medium every day. Colonies started to form in 2–3 weeks and were ready for transfer after an additional week. Undifferentiated colonies were manually picked and transferred to Matrigel-coated six-well plates for expansion. After two rounds of subcloning and expansion (after passage 10), RT-PCR was used to verify whether iPS cells were vector-free with the primer sequences published in the manufacturer's manual.

After iPS cells became virus-free, they were submitted to the University of Minnesota Cytogenomic Laboratory for karyotype analysis. This analysis indicated that the ARPE-19-iPS cells have normal karyotypes.

## Immunofluorescence characterization of ARPE-19-iPS cells

To examine pluripotency markers, iPS cells were fixed with 4% paraformaldehyde for 20 min. If nuclear permeation was required, cells were treated with 0.2% triton-x-100 in phosphate-buffered saline (PBS) for 30 min, blocked in 3% bovine serum albumin in PBS for 2 hr, and incubated with the primary antibody overnight at 4°C. Antibodies targeting the following antigens were used: TRA1-60 (MAB4360, 1:400), TRA1-81 (MAB4381, 1:400), stage-specific embryonic antigen-4 (MAB4304, 1:200), and stage- specific embryonic antigen-3 (MAB-4303, 1:200), all from Millipore/Chemicon (Billerica, MA), OCT3/4 (AB27985, 1:200) from Abcam (Cambridge, MA), and NANOG (EB068601:100) from Everest (Upper Heyford, Oxfordshire, UK). Cells were incubated with secondary Alexa Fluor Series antibodies (all 1:500, Invitrogen) for 1 hr at room temperature and then with DAPI for 10 min. Images were examined using an Olympus FluoView 1000 m IX81 inverted confocal microscope and analyzed with Adobe Photoshop CS6. Direct alkaline phosphatase (AP) activity was analyzed as per the manufacturer's recommendations (Millipore).

## Bisulfite sequencing and methylation analysis

Genomic DNA was isolated using the DNeasy Blood and Tissue kit (Qiagen, Germany) per manufacturer's recommendations for isolation from mammalian cells. Bisulfite conversion was performed using the Epitect Bisulfite kit (Qiagen) according to the manufacturer's protocol for low amounts of DNA. Single-step PCR amplification of the NANOG and OCT4 promoter regions were conducted using Accuprime Supermix II (Invitrogen). Amplification products were visualized by gel electrophoresis and bands were excised and purified using the QIAquick Gel Extraction kit (Qiagen). Purified PCR products were inserted into the PCR4-TOPO vector (Invitrogen) and individual clones were sequenced. Alignment and methylation analysis were performed using the online QUMA program (http://quma.cdb.riken.jp/). Sequenced clones with at least 90% non-CpG cytosine conversion and at least 90% sequence homology were retained for analysis.

## Quantitative reverse transcriptase PCR (*Figure 1—figure supplement 2*)

RNA was isolated using RNeasy Mini kit (Qiagen) and treated with TURBO DNA-free (Ambion, Austin, TX). First-strand cDNA was synthesized using a Superscript III First-Strand Synthesis SuperMix (Invitrogen). Reverse transcriptase-PCR was performed using TaqMan Gene Expression Assays and TaqMan Universal PCR Master Mix, No AmpErase UNG (Applied Biosystems, Carslbad, CA) as per the manufacturer's protocol.

TaqMan gene expression assays used were OCT4 (Hs04260367-gH), SOX2 (Hs01053049-sl), NANOG (Hs04399610-g1), KLF4 (Hs00358836-m1), MYC (Hs00153408-m1), LIN28 (Hs00702808-s1), REXO1 (Hs00810654-m1), ABCG2 (Hs1053790-m1), DNMT3 (Hs00171876-m1), with GAPDH (Hs99999905-m1) used as an endogenous control. Expression levels were measured in duplicate. For genes with expression below the fluorescence threshold, the cycle threshold (Ct) was set at 40 to calculate the relative expression. Analysis was performed using an ABI PRISM 7500 sequence detection system (Applied Biosystems).

## Teratoma analysis

ARPE-19-iPS cells contained in a mixture of DMEM/F12, Matrigel and collagen were implanted onto the hind flank of NSG mice (n = 5) until a palpable mass formed. Teratoma tissue was excised for histological examination following embedding and staining by hematoxylin and eosin. Experiments were conducted with the approval of the Institutional Animal Care and Research Committee at the University of Minnesota.

## Ergodic rate analysis

Ergodic Rate Analysis for cell cycle data was based on previously published work (*Kafri et al., 2013*). First, the raw data from flow cytometry files were extracted using FCSExtract Utility (Earl F Glynn) to comma separated value (.csv) files. The data in the .csv files were then gated in FCS Express 6 (De Novo Software) and the data for only G1-MCM$^{DNA}$ were exported. MCM negative cells were excluded based on the negative control sample (see Flow Cytometry). The mean MCM loading rate was calculated in MATLAB (MathWorks). To calculate the mean MCM loading rate, the G1-MCM$^{DNA}$ were subdivided into 10 equal sized bins, with rate calculated for each bin, and all 10 rates were averaged together for a mean MCM loading rate. The rate calculation was based on the formula from Kafri et al:

$$w_n = \alpha \frac{2-F}{f_n}$$

$w_n$ = MCM loading rate in bin n

$\alpha$ = ln(2)/doubling time (*Figure 2—figure supplement 1*)

F = number of G1-MCM$^{DNA}$ cells/total number of cells in sample. F was calculated from FCS Express and entered into MATLAB manually (*Figure 2—figure supplement 1*).

$f_n$ = number of cells in bin n/number of G1-MCM$^{DNA}$ cells

The bins were created in MATLAB (*Figure 2—figure supplement 1*). To control for small day to day differences in raw data from staining intensities, the histogram edges were defined with the first bin starting at the lowest MCM value and the last bin ending at the highest MCM value, divided into 10 equal sized bins between the lowest and highest MCM value. The 10 $w_n$ were then averaged for a final mean w per sample. Sample MATLAB code:

```
alpha_iPSC = log(2)/15.64;
F_iPSC_1 = 0.0801;
%calculate lowest and highest MCM values%
maxMCM_iPSC_1 = max(iPSC_1(:,1));
minMCM_iPSC_1 = min(iPSC_1(:,1));
%create histogram with 10 bins and specified first and last bin limits%
h10_iPSC_1 = histogram(iPSC_1(:,1), 'NumBins', 10, 'BinLimits', [minMCM_iPSC_1, maxMCM_iPSC_1]);
%calculate fn within each bin%
totalf_iPSC_1 = size(iPSC_1);
fn10_iPSC_1=(h10_iPSC_1.Values)/totalf_iPSC_1(1,1);
%calculate mean w%
w10mean_iPSC_1 = mean(alpha_iPSC. *(2> F_iPSC_1)./fn10_iPSC_1);
```

The mean MCM loading rate was calculated for three biological replicates for each cell line, and the replicates were averaged using GraphPad Prism for further statistical analysis. We cannot use

ergodic rate analysis on actively differentiating cells (e.g. *Figure 3*) because they are not at steady state.

## Quantification and statistical analysis

Statistical analysis was performed with GraphPad Prism seven using unpaired, two-tailed t test (displayed as mean ±SD) or two-tailed Mann-Whitney test as indicated in figure legends. Significance levels were set at $*p \leq 0.05$, $**p \leq 0.01$, $***p \leq 0.001$, $****p \leq 0.0001$. All experiments were performed a minimum of two times, and representative data are shown in figures.

## Acknowledgements

We are grateful to Ran Kafri for advice on ergodic rate analysis. We thank Jeffrey Jones for managerial assistance, Dr. Sam Wolff for microscopy assistance, the UNC Human Pluripotent Stem Cell Core stem cell culture assistance and A Adams, S Wong, M Consuegra, S Goraya, and S Sisk for technical assistance. We also thank Dr. Robert Duronio, Dr. Michael Emanuele, and the Cook lab for helpful discussions. We thank Ron McElmurry and Megan Riddle at the University of Minnesota for conducting teratoma assays, and the Cytogenomics Shared Resource (P30 CA077598) at the University of Minnesota Masonic Cancer Center for karyotype analysis. The UNC Flow Cytometry Core Facility is supported in part by P30 CA016086. This work was supported by a fellowship from the NSF (DGE-1144081) to JPM and by grants from the NIH to JGC (GM083024 and GM102413), RMB (T32-CA009138) JEP (DP2-HD091800) and AKB (GM074917). Additional funding was provided by the WM Keck foundation to JEP and JGC; JT is supported by the Tulloch Chair in Stem Cell Biology, Genetics and Genomics at the University of Minnesota.

## Additional information

### Funding

| Funder | Grant reference number | Author |
| --- | --- | --- |
| National Science Foundation | Graduate Student Research Fellowship DGE1144081 | Jacob Peter Matson |
| National Institutes of Health | Training Grant T32CA009138 | Ryan M Baxley |
| University of Minnesota | Tulloch Chair in Stem Cell Biology, Genetics and Genomics | Jakub Tolar |
| National Institutes of Health | Research Grant GM074917 | Anja-Katrin Bielinsky |
| W. M. Keck Foundation | Research Grant | Jeremy E Purvis Jeanette Gowen Cook |
| National Institutes of Health | Research Grant DP2HD091800 | Jeremy E Purvis |
| National Institutes of Health | Research Grant GM083024 | Jeanette Gowen Cook |
| National Institutes of Health | Research Grant GM102413 | Jeanette Gowen Cook |

The funders had no role in study design, data collection and interpretation, or the decision to submit the work for publication.

### Author contributions

Jacob Peter Matson, Conceptualization, Data curation, Formal analysis, Funding acquisition, Validation, Investigation, Visualization, Methodology, Writing—original draft, Writing—review and editing; Raluca Dumitru, Supervision, Investigation, Methodology; Philip Coryell, Ryan M Baxley, Kirk Twaroski, Beau R Webber, Validation, Investigation, Visualization, Methodology; Weili Chen, Jakub Tolar, Supervision, Validation, Investigation, Visualization, Methodology; Anja-Katrin Bielinsky, Conceptualization, Resources, Supervision, Funding acquisition, Project administration, Writing—review and editing; Jeremy E Purvis, Conceptualization, Resources, Software, Supervision, Funding

acquisition, Writing—original draft, Project administration, Writing—review and editing; Jeanette Gowen Cook, Conceptualization, Resources, Formal analysis, Supervision, Funding acquisition, Visualization, Writing—original draft, Project administration, Writing—review and editing

### Author ORCIDs
Jacob Peter Matson 🄳 http://orcid.org/0000-0002-9375-1676
Jakub Tolar 🄳 http://orcid.org/0000-0002-0957-4380
Anja-Katrin Bielinsky 🄳 https://orcid.org/0000-0003-1783-619X
Jeremy E Purvis 🄳 http://orcid.org/0000-0002-6963-0524
Jeanette Gowen Cook 🄳 https://orcid.org/0000-0003-0849-7405

### Decision letter and Author response
Decision letter https://doi.org/10.7554/eLife.30473.024
Author response https://doi.org/10.7554/eLife.30473.025

## Additional files

### Supplementary files
• Transparent reporting form
DOI: https://doi.org/10.7554/eLife.30473.022

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
