## [Decision Letter]

Thank you for submitting your article "Rapid DNA Replication Origin Licensing Protects Stem Cell Pluripotency" for consideration by *eLife*. Your article has been favorably evaluated by Jessica Tyler as the Senior Editor, Bruce Stillman as the Reviewing Editor, and three reviewers.

The reviewers have discussed the reviews with one another and the Reviewing Editor has drafted this decision to help you prepare a revised submission.

Summary:

The manuscript by Cook and colleagues describes a detailed analysis of the rate of loading MCM2-7 in human cells that have different lengths of G1 phase and different differentiated states. The authors show that there is a clear correlation between the length of G1 and the rate of MCM loading. They show that levels of Cdt1 positively correlate with the length of G1 and that reduction of Cdt1 leads to longer G1 times and reduced rates of Mcm2-7 loading in cells with a short G1. Finally, they show that speeding up G1 by over expression of Cyclin E does not lead to an increased rate of MCM loading, indicating that the length of G1 and MCM loading are not intrinsically linked.

The authors provide strong evidence that the rate of MCM loading is developmentally regulated (most notably in the experiments in which they drive changes in developmental state – either differentiation or de-differentiation). On the other hand, it still remains possible that the two events are not linked per se. It is therefore suggested that a revised paper can be considered if a critical experiment is reported (see Essential revisions).

Overall, although previous studies of the licensing checkpoint may have observed/predicted some of the results in this manuscript, the direct measurement of MCM loading and the ability of modulation of this rate to impact development is important and interesting.

Essential revisions:

It would improve the paper to address whether increased Cdt1 expression in a cell that has a long G1 alters only MCM loading or also shortens G1. This would provide the counterpoint to the Cyclin E experiment and further address whether MCM loading and G1 length are only connected by the licensing checkpoint or in some manner that is not yet understood. That the rate of MCM loading is an important parameter is most clearly demonstrated in Figure 7B, where the authors see that reduced Cdt1 leads to more rapid differentiation but this does not provide evidence of a causative connection.

---

## [Author Response]

Essential revisions:It would improve the paper to address whether increased Cdt1 expression in a cell that has a long G1 alters only MCM loading or also shortens G1. This would provide the counterpoint to the Cyclin E experiment and further address whether MCM loading and G1 length are only connected by the licensing checkpoint or in some manner that is not yet understood. That the rate of MCM loading is an important parameter is most clearly demonstrated in Figure 7B, where the authors see that reduced Cdt1 leads to more rapid differentiation but this does not provide evidence of a causative connection.

The reviewers suggested a complement to analyzing the effects of reducing origin licensing by testing the consequences of overproducing a licensing factor in long G1 cells. The rationale is to further probe the relationship between origin licensing rate and G1 length. We generated a stable derivative of the untransformed (long G1) epithelial cells (RPE_htert) in which Cdt1 is inducibly overproduced. This level of Cdt1 overproduction had little impact on the rate of origin licensing or G1 length in these cells (new Figure 6—figure supplement 1B). This result is perhaps not surprising however because there are many other important licensing proteins that were not overproduced in that experiment (ORC, Cdc6, ORCA, Hbo1, PRSet7/Set8, etc.). Cdt1 is thus necessary but not sufficient for fast origin licensing. Based on previous studies and our own work with untransformed cell lines, we considered Cdc6 to be an alternate candidate for the most rate-limiting licensing protein in this cell line. As discussed below, we had already shown that Cdc6 depletion both slowed licensing and accelerated differentiation in pluripotent stem cells (original Figure 6—figure supplement 1C and 1D and Figure 7—figure supplement 1; in the revision these are Figure 7—figure supplements 2,3). Mammalian Cdc6 is targeted for degradation throughout most of G1 phase by the anaphase promoting complex (Petersen et al. Genes and Dev, 2000), and is only stable in late G1 as a result of Cdk2 activity (Mailand and Diffley, Cell 2005). We generated a stable derivative of RPE cells in which either normal Cdc6 or an APC-resistant Cdc6 mutant are constitutively expressed Figure 6—figure supplement 1A for demonstration that this mutant is stable). We now include as Figure 6E-G, demonstration that the stable Cdc6 mutant is sufficient on its own to induce faster MCM loading. We also presume that this effect will be cell type specific and highly dependent on whichever licensing factor is most rate limiting. Strikingly, Cdc6-induced faster licensing was not enough to substantially shorten G1 in RPE cells – either alone or in combination with Cdt1 overproduction (Figure 6F and Figure 6—figure supplement 1C). Though origin licensing is necessary for S phase onset (in normal human cells, at least), many other events are also required to trigger the G1/S transition. As anticipated by the reviewers, this experiment strengthens the conclusion that origin licensing rate can change independently of G1 length.